# ON THE REPRESENTATION DEGRADATION IN VISION-LANGUAGE-ACTION MODELS

## ABSTRACT

Vision-Language-Action (VLA) models have become a promising paradigm for robotic decision-making, yet their application remains limited by generalization bottlenecks. In this paper, we conduct a layer-wise representation analysis and uncover a previously overlooked phenomenon of representation degradation: deeper layers tasked with action generation exhibit diminished generalization to both semantic information and environmental dynamics. To mitigate this issue, we introduce hidden Space WOrld modeLing (SWOL), a lightweight but efficient approach that aligns degraded deep-layer features with more generalizable mid-layer representations extrapolated from future observations. SWOL enforces temporally consistent, action-grounded representations without modifying model architecture or inference procedures. Extensive experiments in simulation and real-world settings demonstrate that SWOL alleviates representation degradation, leading to improved policy effectiveness and stronger generalization across modalities of vision, language, and dynamics.

## 1 INTRODUCTION

Vision-Language-Action (VLA) models have emerged as a promising paradigm for endowing robots with high-level semantic understanding and low-level motor control through joint training on vision, language, and action data (Brohan et al., 2022; 2023; Octo Model Team et al., 2024; Kim et al., 2024; Black et al., 2024; Kim et al., 2025; Zhao et al., 2025; Li et al., 2025b). By leveraging pre-trained vision-language models (VLMs), VLAs can interpret natural language instructions and generate executable actions in complex environments, enabling generalization across tasks and domains without task-specific engineering (Driess et al., 2023; Team et al., 2025). This integration of perception, cognition, and control has led to impressive performance in diverse robotic manipulation tasks, positioning VLAs as a key step toward general-purpose embodied intelligence. Despite these advances, their application remains limited by generalization bottlenecks.

In this paper, we conduct a layer-wise analysis of fine-tuned VLA models by decomposing their forward pass and evaluating the generalization capability of hidden representations at each depth. We first introduce two evaluation protocols: one to assess **semantics generalization** via task intent classification from instruction-image pairs, and another to measure **dynamics generalization** through inverse dynamics prediction. Our experiments reveal a pervasive phenomenon we term **representation degradation**: while middle layers exhibit strong generalization to unseen language instructions and physical environment, the deeper layers, those responsible for action generation, experience a significant decline in both semantics and dynamics. This degradation creates a critical bottleneck: even if high-quality information is encoded in the earlier layers, it fails to propagate effectively to the decision-making layers, limiting the overall generalization of the policy.

To address this challenge, we draw inspiration from the observation that general agents implicitly encode world models within their internal representations (Richens et al., 2025). We hypothesize that enhancing deep-layer features with predictive ability over latent feature transitions can mitigate representation degradation and improve generalization. Based on this insight, we propose **SWOL** (Hidden Space World Modeling), a simple yet effective objective that encourages the deep-layer perceptual features to predict mid-layer features from the *next* observation. SWOL introduces a lightweight predictor module that maps deep-layer features to reconstruct mid-layer features from

the future time step. This plug-and-play method requires no architectural changes, yet explicitly enhances the dynamics and semantics generalization of VLA representations.

We conduct comprehensive experiments to evaluate SWOL across simulation, real-world tasks, and ablation studies. On the CALVIN benchmark (Mees et al., 2022), SWOL consistently enhances the generalization of three representative VLAs across different data regimes, with particularly pronounced gains in low-data settings. In real-world evaluations on the Aloha platform (Fu et al., 2024) across six manipulation tasks, SWOL improves baseline performance, with the most significant gains observed on tasks requiring fine-grained manipulation, deformable object handling, and complex instruction following. Ablation studies further confirm SWOL's robustness across design choices and hyperparameters. Overall, these results demonstrate that SWOL effectively mitigates representation degradation, strengthens dynamics awareness, and consistently improves the generalization of VLAs.

## 2 REPRESENTATION DEGRADATION IN VLAS

In this section, we identify and characterize a pervasive yet previously undocumented phenomenon in fine-tuned VLA models: **representation degradation**. Despite their critical role in action generation and thus overall policy performance, deeper layers of VLAs exhibit a marked degradation in their ability to generalize across both semantics and dynamics.

Our investigation proceeds in two stages: In Section 2.1, we decompose the VLA forward pass layer by layer to trace how perceptual information propagates and transforms as it informs action representations. In Section 2.2, we introduce and deploy new evaluation protocols to quantitatively assess the generalizability of representations at each layer, separately measuring the fidelity to language semantics and physical dynamics.

### 2.1 FORWARD DECOMPOSITION OF VLAS

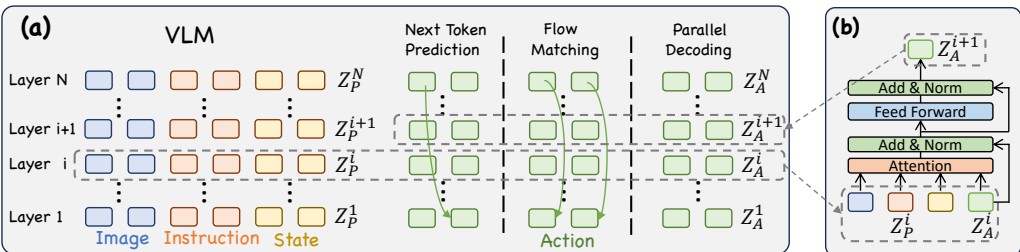

Figure 1: (a) The general architecture of VLAs. (b) Layer-wise forward pass through the VLAs.

We first decompose the forward pass of VLA models from a layer-wise perspective. As illustrated in Figure 1, VLA models take multimodal inputs, including visual observations, language instructions, and proprioceptive states, and process them through an $L$ layer transformer to generate contextualized hidden states and predict an action chunk.

**Layer-wise Update.** We denote the hidden representation at layer $l$ of the VLAs as $Z^l \in \mathbb{R}^{n \times d}$, where $n$ is the sequence length and $d$ is the feature dimension. Following existing works, VLAs are typically structured into two functional components: a perception module that processes visual and language inputs, and an action module that generates actions, allowing us to partition $Z^l$ into $Z^l = \text{cat}(Z_P^l, Z_A^l)$ based on the input token type (Wang et al., 2025; Goyal et al., 2025):

- *Perception representations* $Z_P^l \in \mathbb{R}^{n_1 \times d}$, encoding visual, text, and proprioceptive inputs;
- *Action representations* $Z_A^l \in \mathbb{R}^{n_2 \times d}$, corresponding to the learnable action tokens.

Our analysis focuses on how the representation $Z_A^l$ is updated across layers and affects the output actions. evolves across layers and influences the output actions. We begin by illustrating the layer-wise forward pass in Figure 1. We adopt the decomposition framework proposed in Gandelsman et al. (2024), which yields the following update rule:

$$\hat{Z}_A^l = \text{Attn}\big(Z_P^{l-1}, Z_A^{l-1}\big) + Z_A^{l-1}, \quad Z_A^l = \text{MLP}^l(\hat{Z}_A^l) + \hat{Z}_A^l, \tag{1}$$

$$\text{Attn}\big(Z_P^{l-1}, Z_A^{l-1}\big) = \text{Attn}\Big(P_Q[Z_A^{l-1}], \text{cat}(P_K[Z_P^{l-1}], P_K[Z_A^{l-1}]), \text{cat}(P_V[Z_P^{l-1}], P_V[Z_A^{l-1}])\Big), \quad (2)$$

where $\text{Attn}(\cdot)$ represents the attention module, $P_Q, P_K, P_V$ are the query, key, and value projections, respectively. This attention module is the key operation through which the VLA integrates perceptual information into action generation. MLP denotes the feed-forward network (FFN) applied to refine token representations. By recursively unrolling the layer-wise updates, the cumulative output of the action tokens after a full $L$-layer forward pass can be expressed as:

$$Z_A^L = P\big[Z_A^0\big] + \sum_{l=1}^{L} P\Big[\text{Attn}^l\big(Z_P^{l-1}, Z_A^{l-1}\big)\Big] + \sum_{l=1}^{L} P\Big[\text{MLP}^l\big(\hat{Z}_A^l\big)\Big], \quad (3)$$

where $P[\cdot]$ denotes the linear projection.

**Attention Module Variants.** Different VLAs instantiate the attention mechanism in distinct ways. For instance, $\pi_0$ (Black et al., 2024) employs a flow-matching action decoding scheme combined with block-wise causal self-attention, where tokens within the same modality block can attend to each other as well as to all preceding blocks. $\pi_0$-fast (Pertsch et al., 2025) adopts a next-token prediction decoding scheme while retaining the same attention strategy. OpenVLA-OFT (Kim et al., 2025) enables parallel decoding by replacing the causal self-attention with the bi-directional self-attention.

Despite these architectural variations, equation 3 demonstrates that the output action chunk can be decomposed into a composition of functions acting on layer-wise perceptual representations $Z_P^l$. Consequently, the performance of VLAs is constrained by the quality and generalization capability of these perception-derived features. This highlights the crucial importance of analyzing and enhancing the generalization of the perceptual representations $Z_P^l$ within VLAs.

## 2.2 GENERALIZATION EVALUATION OF LAYER REPRESENTATION

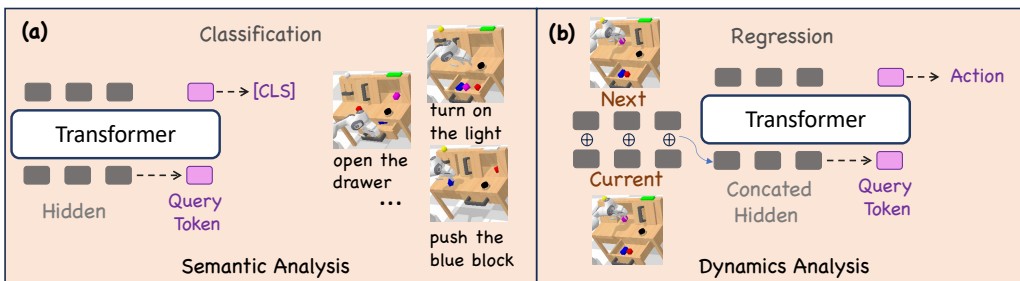

Figure 2: Overview of our analysis framework. (a) Semantics generalization evaluation. (b) Dynamics generalization evaluation.

As shown in equation 3, the output of a VLA can be decomposed into a composition of layer-wise representations. In this section, we experimentally analyze the generalizability of these representations across each layer of the VLA. Unlike large models such as LLMs and VLMs, which primarily focus on perception or text generation, VLAs are policy models that must generate actions aligned simultaneously with linguistic instructions, visual observations, and environmental dynamics. Consequently, representation analysis methods designed for previous large models are ill-suited for VLAs. To address this, we introduce two evaluation metrics, along with their corresponding implementations, to separately assess generalization from the language and the dynamics perspectives.

**Semantics Analysis.** To evaluate how well each layer captures task-relevant linguistic and visual semantics, we adopt a classification framework, illustrated in Figure 2(a). We use the hidden representations $Z_P^l$ from each layer of a fine-tuned model. A lightweight transformer classifier is then trained to predict the semantic task label using features from a dedicated [QUERY] token. The core intuition is that although instructions may vary syntactically or lexically, they often correspond to the same high-level task. By training the classifier to recover this invariant task intent, we probe whether the representation has abstracted away superficial linguistic variation and instead encoded the deeper, task-semantic structure shared across modalities. We evaluate performance on held-out instruction-image pairs to measure how well each layer generalizes to unseen language and visual contexts. Classification accuracy thus serves as a direct indicator of semantic grounding: higher

accuracy implies that the layer's representations are more capable of aligning language and vision into coherent, task-meaningful concepts. The detailed implementation can be found in Appendix C.1.

**Dynamics Analysis.** To probe the dynamics generalization of each layer, we adopt a regression-based inverse dynamics predicting task, depicted in Figure 2(b). Here, we concatenate the hidden representations from the current observation and that of the next observation, and then feed them into a Transformer-based regressor. The model predicts a continuous action vector (e.g., joint torques or end-effector deltas) that maps from the current state to the next state. By training this regressor on a diverse set of trajectories, we can evaluate how well the representation $Z_P^l$ encodes the underlying physical dynamics of the environment. The prediction error on the unseen dataset across layers reflects the *dynamics generalization* of the representation, where lower error indicates better encoding of causal relationships and motion patterns. The detailed implementation can be found in Appendix C.2.

**Setup and Results.** We conduct our representation analysis on the CALVIN benchmark (Mees et al., 2022), evaluating three representative VLA models. All models are first fine-tuned on the same size of expert demonstrations. For *semantics analysis*, we define a classification task over 34 distinct high-level tasks (e.g., "turn on the light", "push into the drawer"). For each task, we sample 10 image–instruction pairs as training data, with diverse instructions describing the same goal. At test time, we evaluate on unseen instructions to measure the model's ability to generalize across language while preserving task semantics. For *dynamics analysis*, we use temporally continuous pairs with all other data settings kept unchanged for training the dynamics regressor, but evaluate on held-out trajectories with novel initial states or object configurations.

The experimental results, summarized in Figure 3, reveal a consistent decline in the generalization capacity of hidden representations across network depth, both for semantics and dynamics. We refer to this phenomenon as **representation degradation** in VLAs. This trend is particularly concerning: although deeper layers are responsible for generating actions and making high-level decisions, they exhibit diminished generalizability to underlying environmental semantics and dynamics. While OpenVLA-OFT shows stable dynamics performance, likely due to bidirectional attention, which encodes action information into perceptual features, we argue that its semantic degradation remains a key bottleneck for generalization.

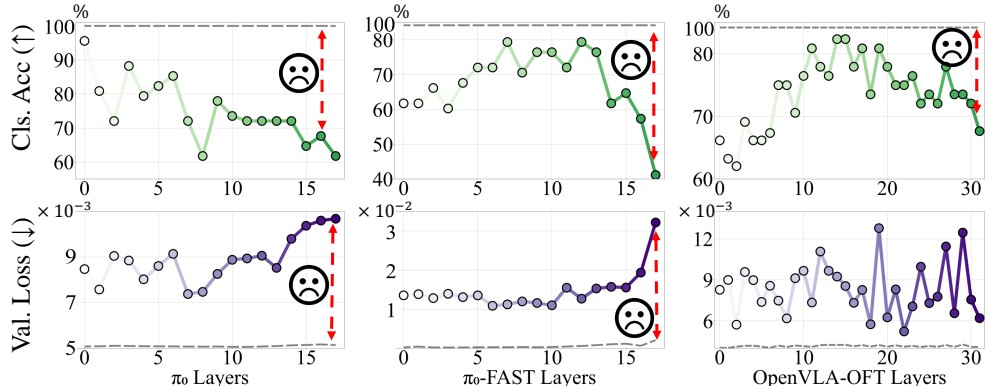

Figure 3: Representation degradation phenomenon in 3 VLAs. Top row: semantic results; Bottom row: dynamics results. Dashed lines denote training performance.

**Represenation degradation harms policy performance**. Moreover, we empirically find that the generalizability in deep layers significantly affects the action generation in such models, making it necessary to address the representation degradation. Specifically, we conduct a layer-wise zero-ablation study following (Nanda et al.; Gandelsman et al., 2024). We replace the representation at the $i-$th hidden layer with zero values during inference, simulating a non-informative or poorly generalizable representation. As shown in Figure 10, this replacement leads to a substantial drop in task success rates, with deeper layers showing the greatest sensitivity. For instance, replacing the representation in the final layer alone causes performance to degrade by more than 20%. These results strongly indicate that hidden representations in VLAs are crucial for shaping robust policy behavior. Thus, preventing representation degradation throughout the network is essential.

## 3 HIDDEN SPACE WORLD MODELING

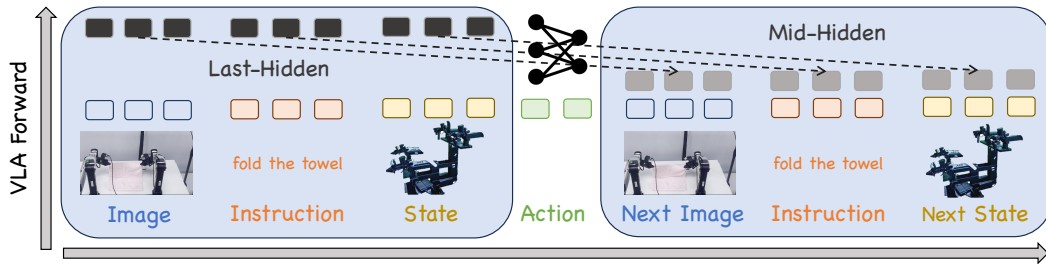

Figure 4: Overview of Hidden Space World Modeling (SWOL). During training, SWOL encourages deep-layer perceptual representations ($Z_P^L$) to predict mid-layer representations ($Z_{P'}^l$) from the next observation, via a lightweight predictor $P_{\text{pred}}$.

### 3.1 KEY INSIGHT

As shown by our analysis in Section 2.2, the generalization of perceptual representations follows a distinctive falling trend across network depth, which may limit the performance of VLAs. One potential solution is to align the degraded features in deep layers with the well-generalized layers. However, due to the presence of residual connections between layers, directly aligning their representations at identical timesteps risks representation collapse.

Fortunately, both theoretical and empirical works suggest a promising alternative: **world modeling**. Previous work (Richens et al., 2025) has shown that any agent achieving a sublinear regret bound over composite goals of depth $n > 1$ must implicitly learn an approximate transition model $\hat{P}(s' \mid s, a)$ of the true environment dynamics $P(s' \mid s, a)$. Specifically, the approximation error satisfies $\left| \hat{P}(s' \mid s, a) - P(s' \mid s, a) \right| = O\left(\frac{\delta}{\sqrt{n}}\right) + O\left(\frac{1}{n}\right)$, where $\delta$ denotes the agent's maximum regret, $n$ is the goal depth, and $P(s' \mid s, a)$ is defined as the exact conditional distribution over future states $s'$ given the current state-action pair $(s, a)$. Therefore, any mechanism that enforces predictability of future states from current state-action pairs is not a heuristic but a **necessary condition** for general agents.

Empirical evidence further reinforces this theoretical foundation. Recent advances in generalist agents demonstrate that world modeling—the capacity to predict environmental dynamics—substantially enhances policy generalization (Hu et al., 2025; Li et al., 2025b; Cen et al., 2025). By forecasting future states based on current observations and actions, agents gain the ability to plan more effectively and maintain temporal coherence across long-horizon tasks. Motivated by this convergence of theory and practice, we propose the following key insight: Representation degradation in deep layers can be mitigated by aligning them with future, better-generalized layers that exhibit stronger predictive capabilities.

Building on this insight, we introduce SWOL, which aligns the features of the deep layer with the *future* mid-layer representations to explicitly equip the degraded representations with world modeling capabilities and improved generalization. This approach offers three key advantages: (1) It improves representation generalization by aligning degraded deep-layer representations with well-generalized mid-layer representations. (2) It implicitly encourages the VLA to better model latent environmental dynamics, which benefits policy generalization. (3) Rather than altering the model architecture or inference procedure, SWOL can be implemented as a self-supervised auxiliary objective without any modification to the existing VLA model.

### 3.2 PRACTICAL IMPLEMENTATION

During training, for each sample at timestep $t$, we first compute the standard forward pass and obtain the final-layer perceptual representation $Z_P^L \in \mathbb{R}^{n \times d}$, where $n$ is the token sequence length and $d$ is the feature dimension. In parallel, we perform an *auxiliary forward pass* using the observation and language instruction from the next action chunk at timestep $t + c$, where $c$ denotes the chunk

length. From this auxiliary pass, we extract the corresponding *mid-layer* perceptual representation $Z_{P'}^l$, which serves as the prediction target.

To connect these two spaces, we introduce a lightweight *predictor* module $P_{\text{pred}} : \mathbb{R}^d \to \mathbb{R}^d$, implemented as a single-layer MLP. The predictor maps each token of $Z_P^L$ to a prediction of its mid-layer counterpart in $Z_{P'}^l$. We optimize this alignment through a token-wise MSE objective:

$$\mathcal{L}_{\text{SWOL}} = \frac{1}{n} \sum_{i=1}^{n} \left\| P_{\text{pred}}(Z_{P,i}^L) - \text{sg}(Z_{P',i}^l) \right\|_2^2, \tag{4}$$

where $n$ is the sequence length of hidden tokens, $\text{sg}(\cdot)$ is the stop-gradient operator, which prevents gradients from propagating back through the auxiliary pass. This ensures that optimization focuses exclusively on enriching the deep-layer features $Z_P^L$, rather than distorting the target $Z_{P'}^l$. By stabilizing the learning dynamics, SWOL preserves the integrity of the mid-layer representations while strengthening the generalization capacity of the deeper layers.

Through this design, SWOL implicitly models transitions in hidden space, enabling deep layers to maintain temporally consistent and action-relevant perceptual states. By counteracting the degradation observed in standard training, SWOL allows VLA models to sustain semantic grounding and dynamics awareness across all layers, leading to stronger generalization for VLA models. Notably, these benefits are obtained without any architectural modifications or inference overhead, making SWOL an attractive and broadly applicable technique for current and future VLA architectures.

## 4 EXPERIMENTS

In this section, we perform comprehensive experiments to answer the following questions: Q1: How does SWOL perform in simulation tasks? (Section 4.1) Q2: How does SWOL perform in real-world tasks? (Section 4.2) Q3: How sensitive is SWOL to hyperparameters and architectural design choices? (Section 4.3) Q4: Can SWOL improve the generalization of representations in VLAs? (Section 4.4))

### 4.1 SIMULATION TASK EXPERIMENTS

| Model | with SWOL | Data | $i^{th}$ Task Success Rate | | | | | Avg. Len ↑ |
|---|---|---|---|---|---|---|---|---|
| | | | 1 | 2 | 3 | 4 | 5 | |
| $\pi_0$ | ✗ | 1% | 0.437 | 0.134 | 0.035 | 0.006 | 0.002 | 0.61 |
| | ✓ | 1% | **0.515** | **0.192** | **0.059** | **0.020** | **0.004** | **0.79** (+29.51%) |
| | ✗ | 10% | 0.914 | 0.760 | 0.626 | 0.494 | 0.380 | 3.17 |
| | ✓ | 10% | **0.940** | **0.806** | **0.666** | **0.539** | **0.432** | **3.38** (+6.62%) |
| | ✗ | 100% | 0.947 | 0.871 | 0.795 | 0.715 | 0.636 | 3.96 |
| | ✓ | 100% | **0.948** | **0.870** | **0.820** | **0.767** | **0.688** | **4.09** (+3.28%) |
| $\pi_0$-fast | ✗ | 1% | 0.058 | 0.001 | **0.000** | **0.000** | **0.000** | 0.06 |
| | ✓ | 1% | **0.100** | **0.013** | **0.000** | **0.000** | **0.000** | **0.11** (+83.33%) |
| | ✗ | 10% | 0.879 | 0.719 | 0.560 | 0.448 | 0.327 | 2.93 |
| | ✓ | 10% | **0.919** | **0.768** | **0.612** | **0.494** | **0.380** | **3.17** (+8.19%) |
| | ✗ | 100% | 0.953 | **0.895** | **0.825** | **0.765** | **0.684** | **4.12** |
| | ✓ | 100% | **0.956** | 0.886 | 0.816 | 0.760 | 0.675 | 4.09 (-0.73%) |
| OpenVLA-OFT | ✗ | 1% | 0.482 | 0.183 | 0.069 | **0.025** | **0.008** | 0.77 |
| | ✓ | 1% | **0.530** | **0.206** | **0.073** | 0.023 | 0.001 | **0.83** (+8.60%) |
| | ✗ | 10% | 0.949 | 0.843 | 0.707 | 0.607 | **0.509** | 3.62 |
| | ✓ | 10% | **0.976** | **0.877** | **0.735** | **0.613** | 0.495 | **3.70** (+2.21%) |
| | ✗ | 100% | 0.983 | 0.907 | 0.847 | 0.758 | 0.675 | 4.17 |
| | ✓ | 100% | **0.992** | **0.948** | **0.879** | **0.799** | **0.709** | **4.33** (+3.87%) |

Table 1: Zero-shot long-horizon performance on the CALVIN benchmark under varying training data regimes. Reported metrics include per-task success rates and average rollout length (Avg. Len).

**Benchmark.** We evaluate SWOL on the CALVIN benchmark (Mees et al., 2022), which is widely used to assess the generalization ability of robotic policies in long-horizon manipulation tasks. In our experiments, we focus on the ABC→D setting, where policies are fine-tuned using different proportions of data(1%, 10%, and 100%) from environments A, B, and C, and evaluated in the unseen environment D. This setup enables us to examine how SWOL performs under varying data regimes.

**Baselines and Evaluation Metrics.** We compare SWOL against three representative VLA policies: (1) $\pi_0$ (Black et al., 2024), pretrained on large-scale real-world robotic datasets and equipped with a flow-matching-based action head, (2) $\pi_0$-fast (Pertsch et al., 2025), which shares the same VLM backbone as $\pi_0$ but replaces the flow-matching head with an autoregressive next-token decoding head for more efficient action generation, and (3) OpenVLA-OFT (Kim et al., 2025), pretrained on the Open X-Embodiment dataset, which integrates pretrained vision and language encoders with parallel action decoding. Performance is measured using the widely adopted metrics of "Success Rate" and "Avg. Len" of completed tasks (higher is better, with values ranging from 0 to 5). The detailed implementation is illustrated in Appendix C.3.

**Task Results.** As shown in Table 1, SWOL consistently enhances the generalizability and long-horizon success of VLA models across varying levels of annotated data. Overall, incorporating SWOL consistently boosts performance across most models and data scales. For $\pi_0$, SWOL yields substantial improvements in low-data regimes, e.g., at 1% data, the average rollout length increases by 29.51%, while still providing gains at 10% (6.62%) and 100% (+3.28%). A similar trend is observed for $\pi_0$-fast, where SWOL notably improves performance at 1% (+83.33%) and 10% (+8.19%), while maintaining comparable performance at 100%. For OpenVLA-OFT, SWOL yields noticeable improvements across all three data regimes (+8.60%, +2.21%, and +3.87%). Especially in 100% data regimes, OpenVLA-OFT with SWOL achieves 4.33 average rollout length, which is a comparatively strong performance for this task. These results reveal that SWOL is effective across varying architectures and data scales (especially in small data regimes), highlighting its robustness and strong generalization capability.

## 4.2 REAL-WORLD TASK EXPERIMENTS

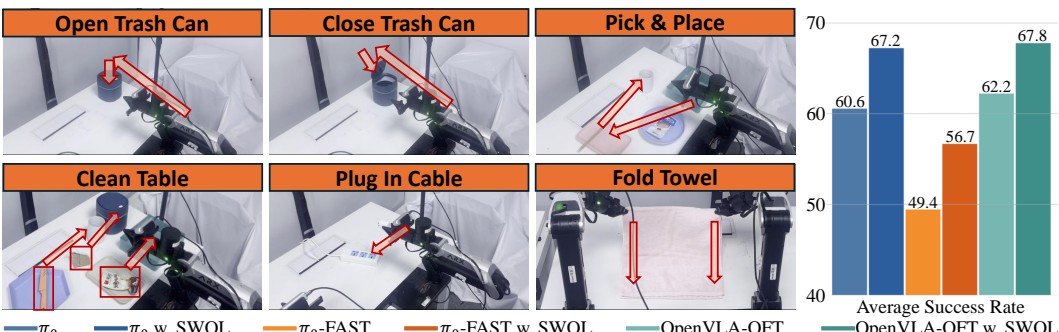

Figure 5: Real-world task illustration and average success rate.

| Model | with SWOL | Open Trash Can | Close Trash Can | Pick & Place | Clean Table | Plug In Cable | Fold Towel |
|---|---|---|---|---|---|---|---|
| $\pi_0$ | ✗ | **28/30** | **27/30** | 20/30 | 7/30 | 4/30 | 23/30 |
| | ✓ | **28/30** | 26/30 | **22/30** | **11/30** | **6/30** | **28/30** |
| $\pi_0$-fast | ✗ | **25/30** | 23/30 | 18/30 | 6/30 | 3/30 | 14/30 |
| | ✓ | 23/30 | **25/30** | **22/30** | **10/30** | **5/30** | **17/30** |
| OpenVLA-OFT | ✗ | 25/30 | 25/30 | 22/30 | 12/30 | 3/30 | **25/30** |
| | ✓ | **28/30** | **26/30** | **25/30** | **15/30** | **4/30** | 24/30 |

Table 2: Performance on real-world tasks. Reported metrics include per-task success rates.

**Task Design.** To evaluate the real-world generalization capabilities of SWOL, we conduct experiments on the ARX5 mobile manipulator platform (see Figure 8), a widely adopted system for training and benchmarking real-world robotic policies. We design a comprehensive suite of tasks that rigorously assess four core generalization abilities: *object generalization*, *spatial generalization*, *instruction following*, and *fine-grained manipulation*. Our evaluation includes 6 manipulation tasks as illustrated in Figure 5. For training, we collect a total of 1030 expert demonstration trajectories using a combination of teleoperation and scripted policies. Each method is evaluated through 180 rollouts—6 tasks with 30 trials per task—to ensure statistically reliable performance comparisons. Further details on the hardware setup, task design, and data collection process can be found in Appendix D.1, Appendix D.2, and Appendix D.3, respectively.

**Task Results.** As shown in Figure 5 and Table 2, in real-world experiments across 6 tasks, we compare three baselines with and without SWOL. For the three relatively simple tasks *open the trash can*, *close the trash can*, and *pick and place*, which primarily require visual and spatial generalization, all approaches achieve high success rates. Nevertheless, our method consistently delivers further improvements across all baselines. For the more challenging tasks, SWOL's improvements are especially meaningful. For *plug in the cable*, which requires precise fine-grained manipulation, $\pi_0$-SWOL achieves the highest success rates, with others performing lower but still capable of solving the task. For *fold the towel*, which demands handling of deformable objects, our method delivers improvements, highlighting its robustness on soft-object manipulation. Notably, $\pi_0$-SWOL achieves near-perfect performance. For *clean the table*, a long-horizon decision-making task, our method provides the most substantial relative boost, suggesting its effectiveness in tasks requiring long-horizon reasoning. Overall, these results demonstrate that SWOL consistently enhances VLA performance across tasks of varying difficulty. While gains on simple tasks confirm its stability in improving fundamental generalization, the more pronounced improvements on fine-grained, long-horizon, and deformable-object tasks highlight SWOL's strength in addressing the key challenges of real-world robotic manipulation.

### 4.3 ABLATION STUDIES

While the main results highlight SWOL's effectiveness, we now explore its robustness across different configurations. We focus on three factors: the *target for representation prediction*, *predictor architecture*, and *world modeling loss weight*. Unless stated otherwise, all experiments use $\pi_0$ on the 10% CALVIN dataset. All ablation results are primarily summarized in Table 3 and detailed in Appendix E

| | Layer 9 Coeffiecient $10^{-4}$ | Target Layer | | Loss Coefficient | | | | | Use TFM |
| --- | --- | --- | --- | --- | --- | --- | --- | --- | --- |
| | | 1 | 13 | $10^{-2}$ | $10^{-3}$ | $10^{-4}$ | $10^{-5}$ | $10^{-6}$ | |
| **Avg. Len** | 3.38 | 3.41 | 3.15 | 2.02 | 3.17 | 3.38 | 3.31 | 3.32 | 3.19 |
| **Relative Change** | +6.62% | +7.57% | -0.63% | -36.28% | +0.00% | +6.62% | +4.42% | +4.73% | +0.63% |

Table 3: Analysis on the target representation prediction layers and loss coefficient for the hidden-space world modeling objectives. We take Layer 9 with coefficient $10^{-4}$ as the default setting of our method. The first row indicates modifications, while the first column donates performance metrics, including the "Avg. Len" of completed tasks and the relative change compared to the baseline.

**Target Representation Prediction:** We examine how the choice of target layer impacts SWOL's performance. Specifically, we predict hidden representations from layers 1, 9, and 13. Results show SWOL is robust to target selection: using layers 1 or 9 improves performance over the baseline, while predicting from layers 5 and 13 yields slightly worse results. Consistent with our earlier analysis, the 1st layer, which encodes rich semantic and spatial information, yields the best performance, with an improvement of up to **+7.57%**.

**Predictor Architecture.** We evaluate the impact of the predictor architecture on robustness. While we used a simple MLP in the main experiments, we also compared it to a Transformer predictor. Both architectures show consistent improvements over the baseline, with the Transformer outperforming the baseline but still lagging behind the MLP. This confirms that the simple MLP is sufficient to leverage world modeling benefits.

**World Modeling Loss Weight:** We analyze the sensitivity of the world modeling loss weight in our method. Varying the loss coefficient shows that SWOL consistently outperforms the baseline, except with very high coefficients. This indicates that the method is broadly effective, with the weight balance influencing performance but not critical to its success.

## 4.4 REPRESENTATION ANALYSIS

Based on the performance gains of SWOL, we aim to examine whether SWOL alleviates the issue of representation degradation. Following the same experimental setup as in Section 2.2, we evaluate baseline models against their SWOL-augmented counterparts.

As shown in Figure 6, SWOL consistently improves both classification accuracy and dynamics modeling in deeper layers—evidenced by the red shaded regions—indicating enhanced semantic and spatial representation quality. Notably, OpenVLA-OFT exhibits minimal dynamics loss regardless of SWOL, suggesting its representations are already robust to degradation. While the characteristic rise-then-fall trend across layers persists, SWOL mitigates the drop-off in deeper layers, preserving generalizable features where they matter most for action generation. We posit that this representational stabilization is the core mechanism behind SWOL's consistent performance gains across diverse data regimes and model architectures.

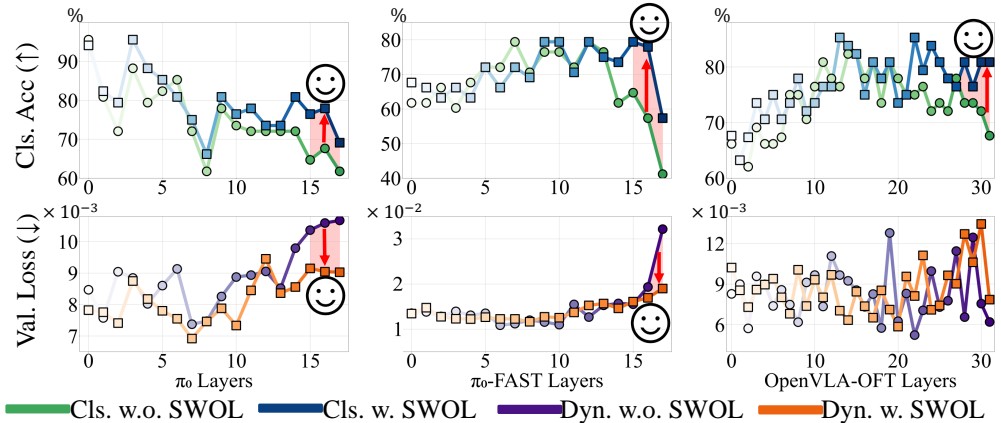

Figure 6: Alleviation of the representation degradation phenomenon.

## 5 RELATED WORKS

**Representation Analysis in Large Models.** Representation analysis has emerged as a prominent tool for understanding large models. In the context of large language models (LLMs), prior works have explored how linguistic structures are encoded, employing methods like probing classifiers (Gurnee & Tegmark, 2023) and information-theoretic measures (Jeon et al., 2024). In vision-language models (VLMs), techniques such as probing (Cao et al., 2020), representational similarity analysis Li et al. (2024), and attention mapping Bi et al. (2025) have been used to explore how semantic concepts and cross-modal alignments emerge at different layers. More recently, some studies have focused on analyzing representations within VLA models, with efforts aimed at preserving pretrained representations (Grover et al., 2025), enhancing latent feature discriminability (Zou et al., 2025), or optimizing representation selection Wang et al. (2025); Reuss et al. (2025). However, these approaches primarily neglect the analysis of generalization in fine-tuned VLA representations, which distinguishes our work.

**Consistency-based Visual Representation Learning.** A wide range of consistency-based objectives has been employed to learn robust visual representations, with a particular focus on temporal consistency. Temporal consistency methods exploit the inherent continuity in videos, including temporal order verification (Misra et al., 2016), cycle consistency across video sequences (Dwibedi et al., 2019), and latent-space future prediction via contrastive objectives (Oord et al., 2018). More recent work blends these ideas by using predictive or cycle-consistency constraints to establish

correspondences across large viewpoint or temporal gaps (Li & Liu, 2023; Baade & Chen, 2025). SWOL significantly differs from these methods by focusing on decision-making in VLAs, while traditional methods target perception tasks. It also addresses representation degradation, a new finding in VLAs, whereas visual representation learning focuses on invariant or predictive features for recognition. Additionally, SWOL introduces a novel hidden-space world modeling loss to enforce consistency between predicted and actual future states in intermediate layers.

**World Modeling for Embodied Control.** Existing research also explores the use of world modeling to enhance the generalization of embodied policies. These methods can be broadly categorized into three types. The first approach utilizes video-based world models to predict future observations or subgoals and then learns inverse dynamics to generate actions (Ye et al.; Du et al., 2023; Black et al., 2023; Chen et al., 2024; Bharadhwaj et al., 2024). The second approach directly uses intermediate representations from video generation models to serve as the perceptual representations for policy models (Hu et al., 2025). Recently, some works have found that incorporating a future prediction loss into the loss function of VLA models can improve their generalization ability (Wu et al., 2023; Cheang et al., 2024; Li et al., 2025a;b; Bu et al., 2025; Zhang et al., 2025a; Cen et al., 2025). However, these methods predict futures directly in raw visual spaces, which is computationally expensive and sensitive to visual perturbations during training. In contrast, our method performs world modeling in the hidden space, which is not only computationally efficient but also can be easily integrated into most VLAs.

## 6    CONCLUSION AND LIMITATIONS

**Conclusion.** In this paper, we identify representation degradation in VLA models: deeper layers lose semantics and dynamics generalization despite their critical role in action generation. To counter this, we propose SWOL, a plug-and-play method that aligns deep features with predictive mid-layer representations of future observations. SWOL improves generalization without changing the model or inference, demonstrating consistent gains in simulation and real-world robotic tasks. The computational resources required for SWOL are detailed in Appendix F.

**Limitations.** There are several limitations in our work. First, it was validated only in imitation learning tasks, so its effectiveness in reinforcement learning (RL) settings remains unexplored. Future work should examine SWOL's impact on RL, where state transition dynamics can vary. Second, we tested SWOL during fine-tuning, but its effectiveness during pretraining is still unknown. Exploring SWOL in pretraining could further enhance generalization. Lastly, while SWOL is efficient and adds no inference overhead, its scalability to larger VLA models and more diverse real-world tasks warrants further investigation.

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

## A  THE USE OF LARGE LANGUAGE MODELS (LLMs)

Large language models (LLMs) were used in the preparation of this manuscript for sentence-level editing, including improving grammar, clarity, and readability.

## B  REPRODUCIBILITY STATEMENT

We commit to ensuring the reproducibility of our research. The source code has been included in the supplementary materials to support replication and independent verification. A comprehensive description of the implementation details is provided in Appendix C.

## C  IMPLEMENTATION DETAILS

### C.1  SEMANTICS GENERALIZATION EVALUATION

**Data Generation.** A total of 340 (34 different tasks and 10 for each task) image-language pairs are collected by randomly sampling from the original Calvin dataset. Then hidden states of these pairs are generated by VLA inference. 80% of the data is used for training, and the remaining 20% is used for testing.

**Training Details.** A Query Decoder Transformer is employed for classification. The input features are first projected into a lower-dimensional embedding space of size 256. A single learnable query token attends to the encoded sequence through one or more transformer decoder layers. Each layer uses 4 attention heads, a feed-forward network with an intermediate dimension of 256, and dropout set to 0.1. After cross-attention and residual updates, the final query representation is passed through a linear classifier to produce logits over 34 semantic categories. The model is trained using cross-entropy loss with Adam optimization (learning rate: $1 \times 10^{-4}$). Performance is assessed at regular intervals over 500 epochs.

### C.2  DYNAMICS GENERALIZATION EVALUATION

**Data Generation.** We use the same number of paired data as used in semantics generalization evaluation for training and testing (34 × 10, 80% train, 20% test). The only difference is that 10 image-language pairs for each task are temporally continuous, as we concatenate hidden states of the current and next timestep to predict the corresponding action.

**Training Details.** A Query Decoder Transformer is used as a regression head. Input features (concatenated from two consecutive hidden states) are projected into an embedding space of dimension 256. A single learnable query token attends to the full sequence through a stack of transformer decoder layers. Each layer uses 4 attention heads, a feed-forward network with an intermediate dimension of 256, and dropout set to 0.1. The resulting query representation is decoded into a 7-dimensional continuous action vector via a linear output layer. The model is trained to minimize MSE loss between predicted and ground-truth actions using Adam ($lr = 1 \times 10^{-4}$). Training runs for up to 200 epochs.

### C.3  BASELINES AND SWOL

$\pi_0$. Table 4 lists hyperparameters for $\pi_0$ fine-tuning. We use full fine-tuning (the default option in their codebase) and train until convergence. The implementation of $\pi_0$ is publicly available at `https://github.com/Physical-Intelligence/openpi`. The implementation of $\pi_0$-FAST can be found in the same code base as $\pi_0$.

$\pi_0$**-FAST.** Table 5 lists hyperparameters for $\pi_0$-FAST fine-tuning. We use full fine-tuning and train until convergence. The hyperparameters are similar to $\pi_0$.

**OpenVLA-OFT.** Hyperparameters for OpenVLA-OFT fine-tuning on our tasks are listed in Table 6. We use LoRA fine-tuning and train until convergence. The implementation of OpenVLA-OFT is publicly available at `https://github.com/moojink/openvla-oft`.

**VLAs with SWOL.** For all experiments except the ablation of using a Transformer-based predictor, we use a single-layer MLP as a hidden predictor. The hidden dimension for input and output is the same, 2048 for $\pi_0$ and $\pi_0$-FAST, 4096 for OpenVLA-OFT.

| hyperparameter | value |
|---|---|
| # GPUs | 4 × NVIDIA A100 (80GB VRAM) |
| learning rate (LR) | 2.5e-5 peak LR (1K steps linear warmup, 29K steps cosine decay to 2.5e-6) |
| total batch size | 64 for simulator tasks and 32 for real-world tasks |
| # train steps | 10k for Calvin 1% and 10% |
| | 30k for Calvin 100% |
| | 10k for real-world tasks |
| input images | 1 third-person camera image, 1 wrist-mounted camera image for Calvin |
| | 1 head camera image, 2 wrist camera images for real-world Fold Towel task |
| | 1 head camera image, 1 right-wrist camera image for other real-world tasks |
| input image size | 224 × 224 px |
| use observation history | no (use single-step inputs) |
| action chunk size | 10 steps for Calvin (predict 10, execute all 10 open-loop at test time) |
| | 20 steps for real-world tasks |
| use proprio (robot state) | no for Calvin, yes for real-world tasks |
| diffusion sampling algorithm | flow matching |

Table 4: $\pi_0$ hyperparameters. This configuration follows the default settings specified in the original $\pi_0$ project codebase.

| hyperparameter | value |
|---|---|
| # GPUs | 4 × NVIDIA A100 (80GB VRAM) |
| learning rate (LR) | 2.5e-5 peak LR (1K steps linear warmup, 29K steps cosine decay to 2.5e-6) |
| total batch size | 64 for simulator tasks and 32 for real-world tasks |
| # train steps | 10k for Calvin 1% and 10% |
| | 30k for Calvin 100% |
| | 10k for real-world tasks |
| input images | 1 third-person camera image, 1 wrist-mounted camera image for Calvin |
| | 1 head camera image, 2 wrist camera images for real-world Fold Towel task |
| | 1 head camera image, 1 right-wrist camera image for other real-world tasks |
| input image size | 224 × 224 px |
| use observation history | no (use single-step inputs) |
| action chunk size | 10 steps for Calvin (predict 10, execute all 10 open-loop at test time) |
| | 20 steps for real-world tasks |
| use proprio (robot state) | no for Calvin, yes for real-world tasks |

Table 5: $\pi_0$-FAST hyperparameters. This configuration follows the default settings specified in the original $\pi_0$ project codebase.

## D  REAL-WORLD TASK SETUP

### D.1  HARDWARE SETUP

Our hardware setup is shown in Figure 8. For embodiment, we use the ARX5 robotic platform, which is similar to Aloha and includes two master arms and two puppet arms. Data are collected via teleoperation, and we use both left and right arms in our experiment. For the vision sensor, a Realsense D435i camera is mounted at the top of the platform to capture RGB image observations.

| hyperparameter | value |
|---|---|
| # GPUs | 4 × NVIDIA A100 (80GB VRAM) |
| learning rate (LR) | 5e-4 |
| total batch size | 32 (8 per GPU) |
| # train steps | 50k for Calvin 1% and 10% |
| | 150k for Calvin 100% |
| | 50k for real-world tasks |
| input images | 1 third-person camera image, 1 wrist-mounted camera image for Calvin |
| | 1 head camera image, 2 wrist camera images for real-world Fold Towel task |
| | 1 head camera image, 1 right-wrist camera image for other real-world tasks |
| input image size | 224 × 224 px for Calvin, 480 × 640 for real-world task |
| use observation history | no (use single-step inputs) |
| LoRA rank | 32 |
| use FiLM | no |
| action chunk size | 10 steps for Calvin |
| | 20 steps for real-world tasks |
| use proprio (robot state) | no for Calvin, yes for real-world tasks |

Table 6: OpenVLA-OFT hyperparameters. This configuration follows the default settings specified in the original OpenVLA-OFT project codebase.

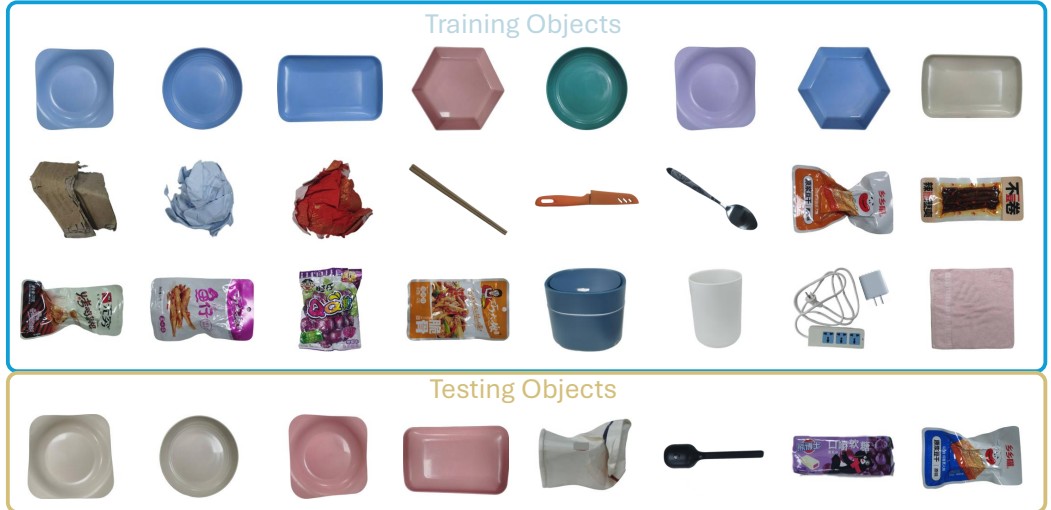

Figure 7: Objects for real-world tasks. All of our real-world experiments include a total of 32 objects. 24 for train and 12 for test.

## D.2 DETAILS OF TASKS

**Open the trash can.** We use a trash can equipped with a spring-loaded switch button. The goal is for the robot to reach the trash can and press the spring button to open the lid. During evaluation, the trash can is randomly placed within a predefined area, exposing the robot to diverse spatial configurations while maintaining task consistency. This task primarily evaluates semantic, visual, and spatial generalization.

**Close the trash can.** This task uses the same trash can as in *Open the trash can*. The goal is to reach the open trash can and press the lid down until it locks securely in place. During evaluation, the trash can is randomly placed within the same predefined area as in *Open the trash can*. This task also focuses on semantic, visual, and spatial generalization.

**Pick and Place.** In this task, a variety of objects are placed on the table, and the robot is instructed to *pick X in Y and place it into Z*. Here, **X** denotes the target object, which may be *cutlery*, *snack*, or *trash*; **Z** denotes the target receptacle, including *cup*, *box*, or *trash can*; and **Y** denotes the initial

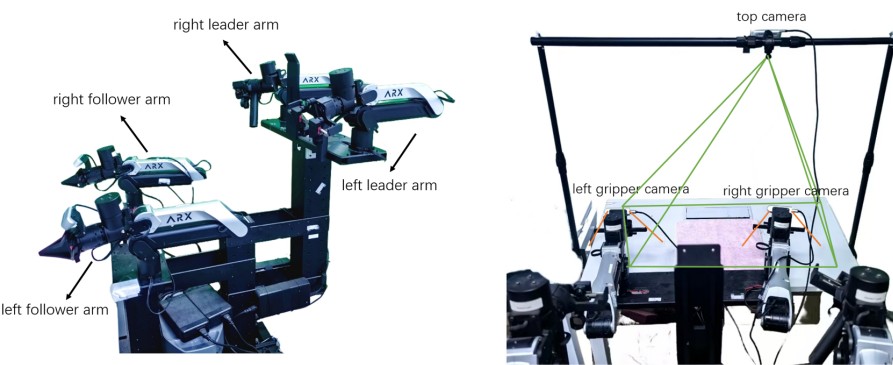

Figure 8: Hardware illustration.

location of the object, either *on the table* or *in a [color] [shape] plate*. Colors include *blue, green, pink, brown,* and *purple*, while shapes include *round, rectangular, square,* and *hexagonal*. In total, there are 3 object categories, 3 possible target receptacles, and 13 different initial placements. We use 8 snack items, 4 pieces of cutlery, and 4 types of trash objects, with 6 snacks, 3 cutlery items, 3 trash items, and 8 plates allocated for training; the remaining items are reserved for evaluation. During evaluation, test objects are randomly sampled and placed within predefined areas. This task evaluates object, semantic, visual, and spatial generalization.

**Clean the table.** This is a long-horizon decision-making task composed of 3–7 sequential subtasks. Subtasks are drawn from the three tasks described above, along with an additional task, **Push plates**. In **Push plates**, the instruction is *put all plates in a straight line*, requiring the robot to push 2–3 plates accordingly. Task configurations follow the same setup as in **Pick and Place**. During evaluation, test objects are randomly sampled and placed within predefined areas. Importantly, evaluation is interactive: in each episode, a new instruction is issued once the robot completes the current subtask. In addition to the generalization abilities tested in **Pick and Place**, this task further challenges the robot's long-horizon decision-making and multi-task comprehension abilities.

**Plug in the cable.** In this task, the robot must insert a plug into its corresponding socket and tap the plug to ensure it is fully secured. During evaluation, both the plug and the socket are randomly placed within predefined areas. While this task is relatively simple in terms of semantics, object, and visual generalization, it poses greater challenges in spatial generalization and fine-grained manipulation.

**Fold the towel.** Here, the robot's objective is to fold a towel twice, reducing it to one-quarter of its original size. The folding process requires step-by-step execution and careful adjustments in positioning. During evaluation, the initial towel position is perturbed with only minor random noise. This task primarily evaluates the ability to manipulate deformable objects as well as long-horizon decision-making.

### D.3 DATA COLLECTION

We collect real-world data with ARX5 robotic platform manually. Totally, 1030 expert demonstration trajectories are collected for training. The detailed # of trajectories for each task is shown in Table 7. The demonstrations for each task are shown in Figure 9.

## E ADDITIONAL EXPERIMENTAL RESULTS

In this subsection, we present the detailed performance of ablation studies. All subsequent experiments are conducted using the $\pi_0$ model with 10% data.

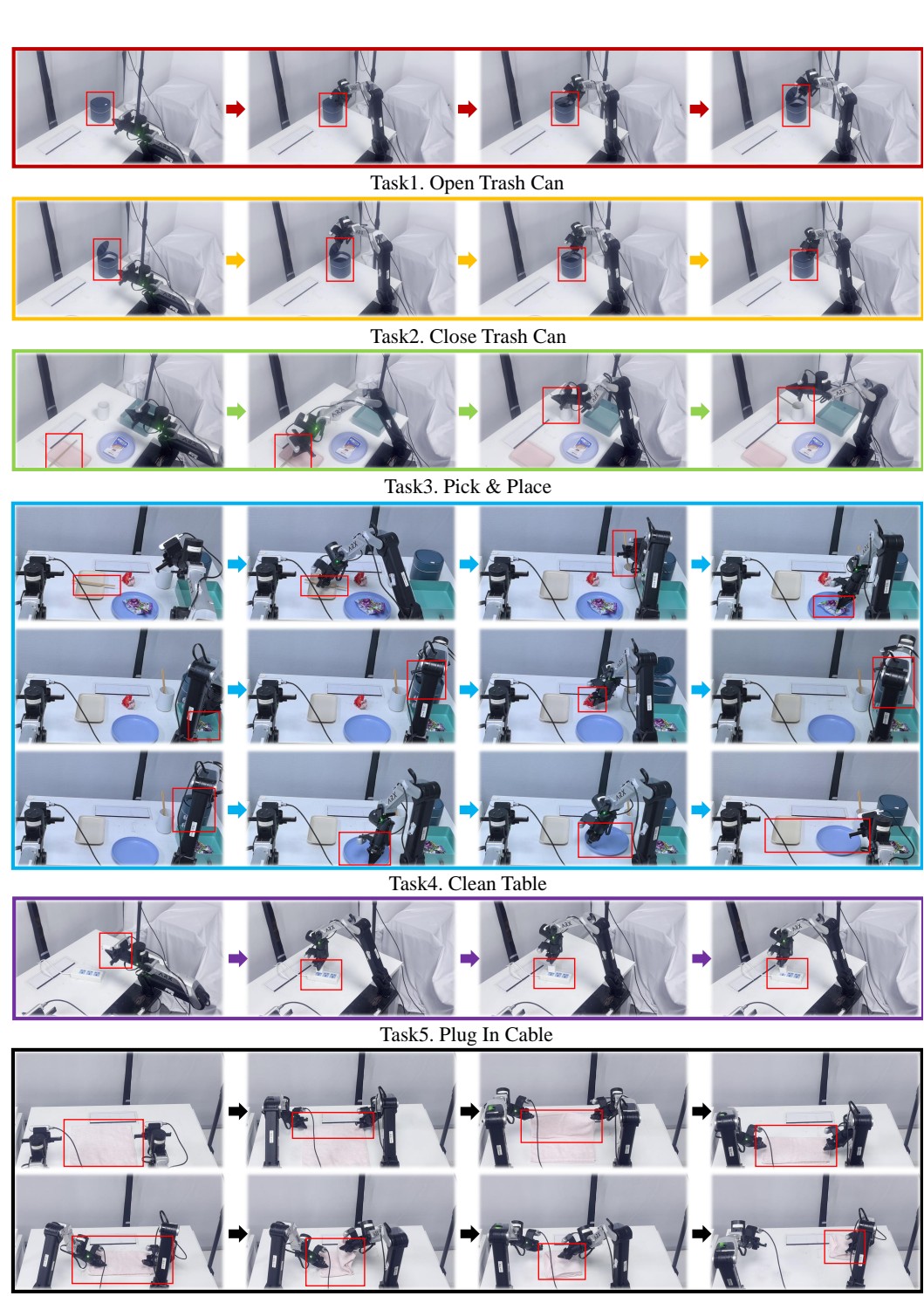

Figure 9: Demonstrations for real-world tasks.

| Task | # of traj. |
|---|---|
| Open Trash Can | 120 |
| Close Trash Can | 120 |
| Pick & Place | 540 |
| Clean Table | 180 |
| Plug In Cable | 50 |
| Fold Towel | 20 |

Table 7: # of expert demonstration trajectories for each task.

## E.1 TARGET REPRESENTATION PREDICTION

Detailed results on how the choice of target layer impacts SWOL's performance are presented in Table 8. We compare the baseline with SWOL using target layers 1, 5, 9, and 13.

| Model | Data | Target Layer | $i^{th}$ Task Success Rate | | | | | Avg. Len ↑ |
|---|---|---|---|---|---|---|---|---|
| | | | 1 | 2 | 3 | 4 | 5 | |
| $\pi_0$ | 10% | — | 0.914 | 0.760 | 0.626 | 0.494 | 0.380 | 3.17 |
| | | 1 | **0.946** | **0.821** | 0.663 | **0.547** | **0.437** | **3.41** (+7.57%) |
| | | 5 | 0.913 | 0.769 | 0.610 | 0.473 | 0.368 | 3.13 (-1.26%) |
| | | 9 | 0.940 | 0.806 | **0.666** | 0.539 | 0.432 | 3.38 (+6.62%) |
| | | 13 | 0.903 | 0.756 | 0.606 | 0.494 | 0.395 | 3.15 (-0.63%) |

Table 8: Analysis on the target representation prediction layers.

## E.2 PREDICTOR ARCHITECTURE

Detailed results on how predictor architecture impacts SWOL's performance are presented in Table 9. We compare the baseline with SWOL using MLP and Transformer as predictor.

| Model | Data | Predictor Architecture | $i^{th}$ Task Success Rate | | | | | Avg. Len ↑ |
|---|---|---|---|---|---|---|---|---|
| | | | 1 | 2 | 3 | 4 | 5 | |
| $\pi_0$ | 10% | — | 0.914 | 0.760 | 0.626 | 0.494 | 0.380 | 3.17 |
| | | MLP | **0.940** | **0.806** | **0.666** | **0.539** | **0.432** | **3.38** (+6.62%) |
| | | Transformer | 0.916 | 0.771 | 0.630 | 0.495 | 0.382 | 3.19 (+0.63%) |

Table 9: Analysis on the architecture for the hidden-space world modeling predictor.

## E.3 WORLD MODELING LOSS WEIGHT

Detailed results on how world modeling loss weight impacts SWOL's performance are presented in Table 10. We compare the baseline with SWOL using different values of loss coefficient.

## E.4 LAYER-WISE ZERO-ABLATION

We conduct a layer-wise zero-ablation study with $\pi_0$ model finetuned on 10% CALVIN dataset to validate the effectiveness of the representation in each layer empirically. Specifically, we replace the representation at the $i-$th hidden layer with the zero values, simulating a non-informative or poorly generalizable representation. As shown in Figure 10, this replacement leads to a substantial drop in task success rates, with the deeper layers showing the greatest sensitivity. These results indicate that hidden representations in VLAs are crucial for shaping robust policy behavior. Thus, preventing the representation degradation of VLAs is essential.

| Model | Data | Coefficient | $i^{th}$ Task Success Rate | | | | | Avg. Len $\uparrow$ |
|---|---|---|---|---|---|---|---|---|
| | | | **1** | **2** | **3** | **4** | **5** | |
| $\pi_0$ | 10% | 0 | 0.914 | 0.760 | 0.626 | 0.494 | 0.380 | 3.17 |
| | | $10^{-2}$ | 0.758 | 0.507 | 0.347 | 0.236 | 0.168 | 2.02 (-36.28%) |
| | | $10^{-3}$ | 0.893 | 0.762 | 0.620 | 0.497 | 0.397 | 3.17 (+0.00%) |
| | | $10^{-4}$ | **0.940** | **0.806** | **0.666** | **0.539** | **0.432** | **3.38** (+6.62%) |
| | | $10^{-5}$ | 0.925 | 0.805 | 0.645 | 0.524 | 0.414 | 3.31 (+4.42%) |
| | | $10^{-6}$ | 0.927 | 0.811 | 0.648 | 0.521 | 0.409 | 3.32 (+4.73%) |

Table 10: Analysis on the loss coefficient for the hidden-space world modeling objectives

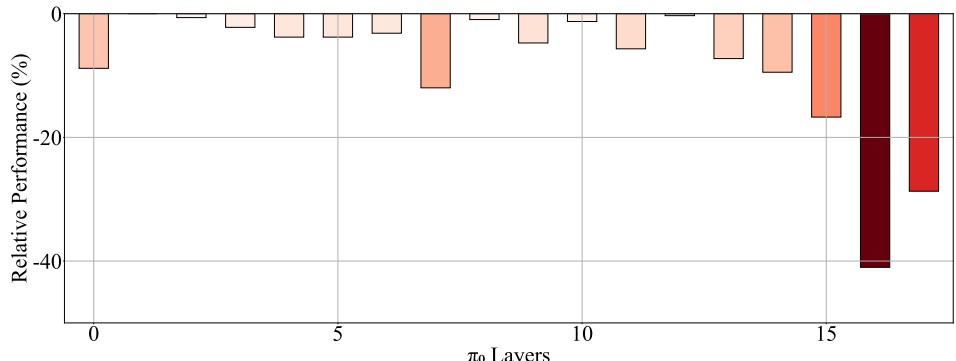

Figure 10: Layer-wise zero-ablation results. The relative performance refers to the change in final performance on the CALVIN benchmark after ablating the i-th layer.

### E.5 Comparison Results with Additional Baselines

We introduce 3 additional world modeling baselines on top of the $\pi_0$ architecture: (1) Direct World Modeling (DWM). In this variant, we apply the world modeling loss using the first-layer representation as input, ensuring that the dynamics loss does not propagate to or regularize the degraded hidden-layer representations. (2) Direct World Modeling with Degraded layers (DWMD). The input and target layer of world modeling loss are both the last layer (degraded hidden layer) representation. (3) CoT-VLA. We extend the input sequence by appending future image tokens after the current image and language tokens, and introduce an additional image decoder head trained with a world modeling loss that predicts these future frames in pixel space.

We train these models on the 10% CALVIN dataset using different modeling methods. All methods achieve a nearly identical final imitation loss(0.05557 for $\pi_0$, 0.05564 for $\pi_0$-CoT-VLA, 0.05566 for $\pi_0$-DWM, 0.05607 for $\pi_0$-DWMD, and 0.05534 for $\pi_0$-SWOL), after 10,000 training steps. However, their test policy performance differs noticeably as shown in Table 11, with a clear ranking: $\pi_0$-SWOL > $\pi_0$-DWMD > $\pi_0$-DWM > $\pi_0$ > $\pi_0$-CoT-VLA.

We include DWM and DWMD as baselines to represent methods that entirely ignore layer generalization and degradation and focuses solely on world modeling. The difference in DWMD from DWM is that the loss is applied directly to the degraded layer. The superior performance of SWOL against DWM and DWMD suggests that its benefits stem critically from the ability to mitigate representation degradation, not the world modeling objective itself. DWMD performs slightly better than DWM but leads to highly unstable training, with frequent loss spikes. Our experiments demonstrate that naively applying CoT-VLA to off-the-shelf VLA models can actually degrade generalization. This may stem from the fact that its original implementation uses the dynamics loss during pretraining, while our version applies it only during fine-tuning.

| Model | Modeling Method | $i^{th}$ Task Success Rate | | | | | Avg. Len ↑ |
|---|---|---|---|---|---|---|---|
| | | **1** | **2** | **3** | **4** | **5** | |
| $\pi_0$ | $\pi_0$-Baseline | 0.914 | 0.760 | 0.626 | 0.494 | 0.380 | 3.17 |
| | DWM | 0.918 | 0.765 | 0.626 | 0.504 | 0.412 | 3.23 (+1.89%) |
| | DWMD | 0.924 | 0.791 | 0.626 | 0.516 | 0.405 | 3.26 (+2.84%) |
| | CoT-VLA | 0.880 | 0.725 | 0.565 | 0.430 | 0.341 | 2.94 (-7.26%) |
| | SWOL | **0.940** | **0.806** | **0.666** | **0.539** | **0.432** | **3.38** (+6.62%) |

Table 11: Comparison performance on the CALVIN benchmark with world modeling baselines.

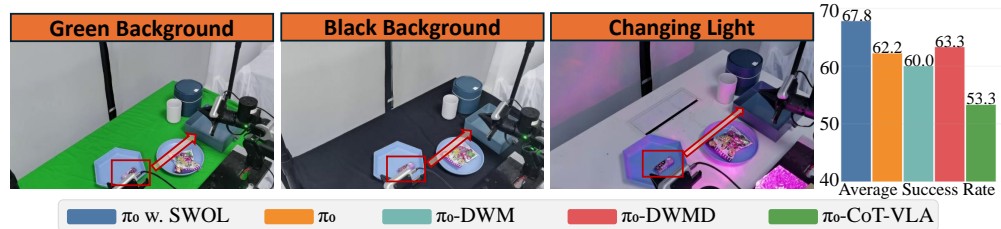

Figure 11: Illustration of Real-world tasks with environmental clutter and average success rate.

## E.6   ADDITIONAL READ-WORLD TASK EXPERIMENTS

To further validate the visual robustness of SWOL, we conduct more challenging real-robot experiments with increased background clutter, as shown in Figure 11. These experiments are performed on the Pick&Place task, we design 3 more challenging scenes considering background clutter and lighting conditions: (1) Black table background. (2) Green table background. (3) Color-changing lights. For all models, the same real-world dataset in 4.2 is used for training. This indicates that the test scenarios involve highly unseen and heavily perturbed environments. We follow the same evaluation configuration in 4.2 and evaluate 30 trials per scene. The results are shown in Table 12. Performance follows a ranking of $\pi_0$-SWOL > $\pi_0$-DWMD > $\pi_0$ > $\pi_0$-DWM > $\pi_0$-CoT-VLA. SWOL exhibits the strongest robustness to environmental disturbances.

| Model | Modeling Method | Black Background | Green Background | Changing Light | Avg. Success Rate |
|---|---|---|---|---|---|
| $\pi_0$ | $\pi_0$-Baseline | 19/30 | 19/30 | **18/30** | 62.2 |
| | DWM | 18/30 | 19/30 | 17/30 | 60.0 |
| | DWMD | 20/30 | 20/30 | 17/30 | 63.3 |
| | CoT-VLA | 16/30 | 17/30 | 15/30 | 53.3 |
| | SWOL | **22/30** | **21/30** | **18/30** | **67.8** |

Table 12: Performance on real-world tasks with environmental clutter. Reported metrics include per-task and average success rates.

## E.7   LAYER REPRESENTATION EVALUATION OF PRETRAINED VLA AND VLM

The current VLA training process can be broadly divided into three stages: (1) Pretrained VLM. (2) Pretrain VLA on Large-Scale, Diverse Robot Data. (3) Finetune VLA on Task-Specific Robot Data. We analyze the models at each of these stages for $\pi_0$-FAST, a discrete action decoding VLA model, with the results presented in Figure 12. The results of $\pi_0$-FAST and $\pi_0$-FAST-SWOL is the same with in 4.4. Using exactly the same experimental protocol, we add semantics and dynamics analysis of Paligemma(the VLM backbone of $\pi_0$-FAST) and pretrained $\pi_0$-FAST. Our experimental analysis reveals distinct trends in the representation layers of these models.

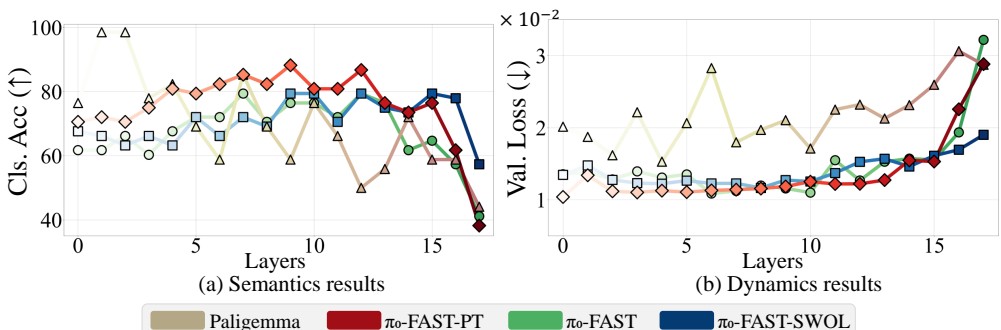

Figure 12: Semantics and dynamics analysis results on pretrained models. The analysis examines classification accuracy and dynamics loss compared with finetuned models with and without SWOL.

Based on these findings, we draw the following conclusions: (1) Pretraining on robot data reduces the semantic generalization of VLM. (2) Pretraining on robot data enhances the dynamic generalization of VLM. (3) All models exhibit varying degrees of representation degradation, with the degradation being more pronounced after pretraining and finetuning on robot data. However, SWOL significantly alleviates this degradation. The underlying reasons for the first two conclusions may lie in the fact that, during VLM pretraining, the model has not been exposed to a large volume of robot data. Consequently, when VLA is trained on data with a significantly different distribution (robot data), it disrupts some of the semantic representations within the VLM while simultaneously boosting its ability to generalize dynamic behaviors.

### E.8 ADDITIONAL REINFORCEMENT LEARNING EXPERIMENT

We include a new RL experiment on the Libero Spatial benchmark (Liu et al., 2023), a widely used benchmark for VLA-RL. Specifically, we begin with a 5-shot SFT $\pi_0$ model and perform RL via Reinflow (Zhang et al., 2025b; Chen et al., 2025) with LoRA (Hu et al., 2022). The detailed hyperparameters are listed in Table 13. During training, we incorporate a hidden-space world modeling loss with $1e-5$ weight analogous to SWOL. Our results show that SWOL achieves a best success rate of **92.2%**, compared to 85.9% for the baseline (w.o. SWOL), demonstrating that our approach effectively generalizes to RL settings. Full training curves are provided in Figure 13.

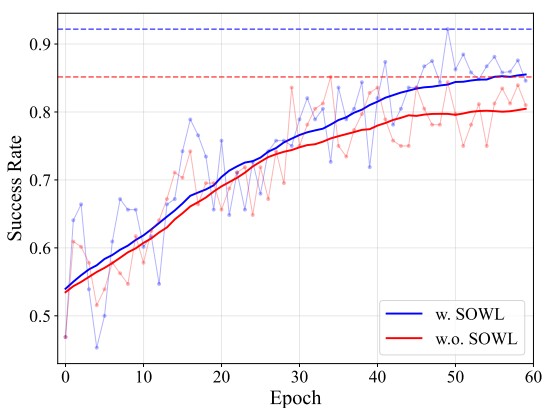

Figure 13: Visual comparison of RL experiments.

| Parameters | |
|---|---|
| Batch size | 64 |
| Actor lr | 1e−5 |
| Critic lr | 1e−4 |
| Lora rank | 32 |
| Reward discount rate $\gamma$ | 0.99 |
| GAE $\lambda$ | 0.95 |
| Clip ratio $\epsilon$ | 0.2 |
| Rollout nums | 512 |
| Action chunk $H$ | 10 |
| Denoise steps | 4 |
| Noise max log-var | 0.16 |
| Noise min log-var | 0.08 |
| Entropy bonus | 0.005 |

Table 13: Hyperparameters for RL.

### E.9 QUALITATIVE VISUALIZATION OF REPRESENTATION STRUCTURE

In this section, we visualize how SWOL reshapes representations to better understand its impact on VLA features. Since SWOL enforces strong alignment across time, we conduct a qualitative

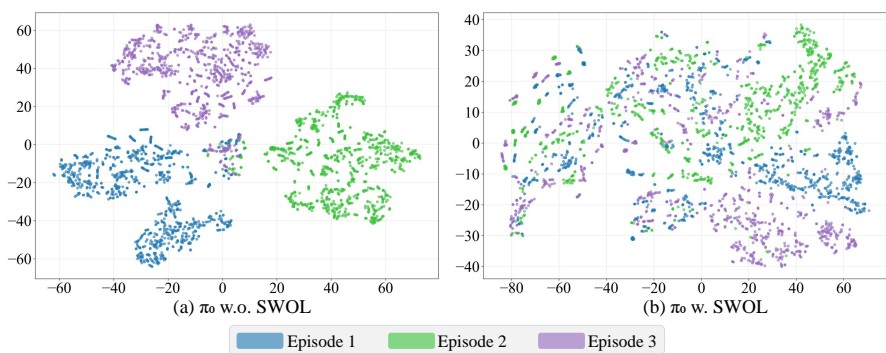

Figure 14: t-SNE results on time-variant data.

analysis on time-continuous data. We select three episodes from different tasks, each consisting of 10 consecutive steps. For each step, we generate hidden representations as in Section 2.2 and visualize 100 tokens per step. As shown in Figure 14, the representations of a single episode become more spatially dispersed after applying SWOL, indicating improved diversity in action representations. This observation is further validated by quantitative metrics: the mean centroid distance increases from 23.63 to **46.46**, and the convex hull area grows from 219.73 to **385.23**, confirming consistent diversity improvement of SWOL.

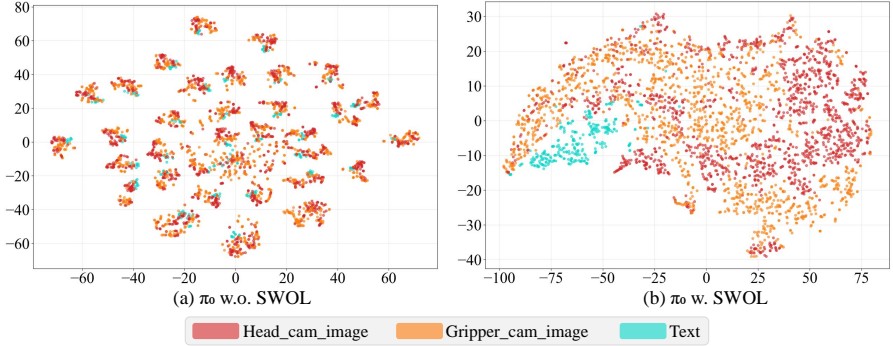

Figure 15: t-SNE results on semantics data.

To further understand how SWOL changes the representations of VLAs, we randomly sample image–text pairs from each of 34 distinct tasks and generate hidden representations following the procedure in Section 2.2. Using t-SNE (Maaten & Hinton, 2008), we visualize a total of 3,400 tokens. As shown in Figure 15, representations before applying SWOL are intermixed across third-person camera images, gripper camera images, and text instructions. In contrast, after applying SWOL, the representations exhibit clear separation across modalities.

## F  COMPUTATIONAL RESOURCES

All models with and without SWOL are run on 4 × NVIDIA A100 (80GB VRAM). The detailed GPU hours for different VLAs and tasks are present in Table 14. The increased computational cost primarily stems from the additional forward pass required to compute mid-layer features of future states. However, since SWOL does not modify the VLA's architecture or inference procedure, it introduces no extra computation at inference time.

| Model | # of GPU hours |
|---|---|
| $\pi_0$ w.o. SWOL | around 20 for 1%, 10% Calvin, around 60 for 100% Calvin around 20 for all real-world tasks |
| $\pi_0$ w. SWOL | around 26 for 1%, 10% Calvin, around 78 for 100% Calvin around 26 for all real-world tasks |
| $\pi_0$-FAST w.o. SWOL | around 28 for 1%, 10% Calvin, around 84 for 100% Calvin around 28 for all real-world tasks |
| $\pi_0$-FAST w. SWOL | around 36 for 1%, 10% Calvin, around 108 for 100% Calvin around 36 for all real-world tasks |
| OpenVLA-OFT w.o. SWOL | around 106 for 1%, 10% Calvin, around 318 for 100% Calvin around 106 for all real-world tasks |
| OpenVLA-OFT w. SWOL | around 138 for 1%, 10% Calvin, around 414 for 100% Calvin around 138 for all real-world tasks |

Table 14: Computational resources for different VLAs and tasks.

