# OpenReview forum: "On the Representation Degradation in Vision-Language-Action Models"
_ICLR.cc/2026/Conference — Submitted to ICLR 2026_

### Official Review · Reviewer_by5U · 2025-10-20

**Soundness:** 3
**Presentation:** 3
**Contribution:** 2
**Rating:** 4
**Confidence:** 4

**Summary:**

The work isolates the issue of representation degradation within vision-language action models, where deeper layer fail to carry rich information (semantic and dynamical) that is useful for generalization. These metrics are defined in this context and three VLA models are analyzed to exhibit these shortcomings. The solution proposed to this issue, SWOL, consists in encouraging the deeper perceptual features to match mid-level features from the next observation in time. This is done as to force some sort of world model representation learning at those deeper layers. This is evaluated in simulation as well as in the real world and some performance benefits are shown.

**Strengths:**

The paper topic is very relevant to the current approaches to robot learning, and the problem highlighted is clearly identified.

The metrics to evaluate the degradation phenomenon are clear and the case made for this being an issue is sound and convincing.

The design with the self-supervised loss aligning representations between timesteps is quite elegant.

The experiments run are adequate to test the hypothesis made and solution suggested.

The performance gains, though not miraculous, are sufficiently beneficial for this to be an interesting result.

**Weaknesses:**

**Insight and implementation**
- The authors resort to world modeling as the general idea behind the solution for mitigating representation degradation in deep VLA layers. Just in terms of presentation, the insight should not be a question (lines 231-232) rather an observation.
- The insight on its own is insufficient to directly lead to the solution proposed: current deep layer features aligned with future mid layer features. This warrants more explanation as to why this specifically is the best way and not just one way to do things.
- The mid level representations seems to also vary substantially in quality both within the same architecture (e.g. pi_0's 8th layer does very poorly on semantic classification) and among VLAs (Very noisy for openVLA while very clean for pi_0-Fast). Admitting that the observation is general and empirical, this is still not discussed to a sufficient extent, and the solution does not seem to directly account for this.
- The presentation of the partitioning of the hidden features into perceptual and action parts is not clearly presented. It is unclear to the reader why such a partitioning is taken as a postulate. The authors cite Gandelsman et al., 2024, but the text is in no way self-contained in presenting this decomposition method and relating it to the policy architectures considered.
- Along these lines, the update rules (lines 112-124) are hard to decipher both because of the above point, but also due to the cumbersome notation. I would suggest rewriting this section as well as creating a more technical figure that exhibits the decomposition and update rules in a clear fashion, even should it be in the appendix.

**Presentation**
- I did not find figure 1 very useful, especially considering the area/information ratio.
- In table 1 the best per data fraction and per task score should be in bold as it is fatiguing on the eyes to decipher such a big block of numbers. This is done for the average length but I suspect practitioners care more about success rate.
- The paper fails to report a reproducibility statement as well as a statement on the use of LLMs which I believe are required

**Questions:**

- Why does it make sense to do deep-layer to mid-layer of next step matching?

- What is the "target" mid-layer selection protocol?

- Why is the average length an interesting/meaningful metric (table 1) ?

---

> ### Author Response · Authors · 2025-11-21
>
> We sincerely appreciate the reviewer’s constructive feedback and would like to offer further clarification in response.
>
> >**W1**: The authors resort to world modeling as the general idea behind the solution for mitigating representation degradation in deep VLA layers. Just in terms of presentation, the insight should not be a question rather an observation.
>
> Thank you for raising this point. We have revised it into a clear declarative statement as "We can mitigate the representation degradation via aligning degraded hidden layers with future well-generalized layers.".
>
> >**W2, Q1**: The insight on its own is insufficient to directly lead to the solution proposed: current deep layer features aligned with future mid-layer features. This warrants more explanation as to why this specifically is the best way and not just one way to do things.
>
> **TL;DR: We clarify our design choice from two perspectives, (i) the theoretical motivation for using future features as prediction targets, and (ii) the empirical stability of mid-layer features as robust representations. We further add two baselines to emprically validate the effectiveness and superiority of our design choice.**
>
> We appreciate the reviewer's valuable comment and would like to provide clarification. We analyze the rationale behind using current deep-layer features aligned with future mid-layer features to mitigate representation degradation from two complementary perspectives: **(i) its reliance on predicting future features as targets, and (ii) its use of mid-layer features as predictive representations**.
>
> 1. **The role of predicting future features.** Previous work [1] has established that any agent satisfying a regret bound over composite goals of depth $n > 1$ **must** implicitly encode an approximation transition $\hat{P}(s' \mid s, a)$ of the true environment transition function $P(s' \mid s, a)$, with error $\bigl| \hat{P}(s' \mid s, a) - P(s' \mid s, a) \bigr| = \mathcal{O}\left(\frac{\delta}{\sqrt{n}}\right) + \mathcal{O}\left(\frac{1}{n}\right),$ where $\delta$ is the agent’s maximum regret and $n$ is the goal depth. Crucially, the transition function $P(s' \mid s, a)$ is defined precisely as the conditional distribution over future states $s'$ given current state $s$ and action $a$. Therefore, any mechanism that enforces predictability of future states from current state-action pairs is not a heuristic but a **necessary condition** for general agents. Motivated by this theorem, SWOL’s auxiliary loss, which minimizes the distance between predicted and actual future representations, explicitly enforces this condition, thereby ensuring that the agent’s policy satisfies the preconditions under which a world model is provably recoverable.
> 2. **The role of predicting mid-layer features.** In this paper, the core issue we aim to address is the empirical phenomenon of representation degradation in VLAs. The reason we use mid-layer as the target is that we observe that **mid-layer representations across different VLAs are consistently robust**. **To reduce tuning overhead when applying SWOL to various VLAs**, we default to using a mid-layer as the target for practicality and stability.
>
> Moreover, to further validate the effectiveness and superiority of our method, we add two world-modeling baselines:
>
> 1. **Direct World Modeling (DWM).** We apply the world modeling loss using the first-layer representation as input, ensuring that the dynamics loss **does not influence or regularize the degraded hidden-layer representations.**
> 2. **CoT-VLA** [2]: Following the original formulation, we reimplement Cot-VLA on top of the $\pi_0$ architecture to ensure a fair comparison. We extend the input sequence by appending future image tokens after the current image and language tokens, and introduce an additional image decoder head trained with a world modeling loss that **predicts these future frames directly in pixel space**.
>
> The results are shown below, demonstrating that SWOL performs consistently better than all other baselines. The limited improvement from DWM suggests that merely adding a predictive loss on image features is insufficient; the key benefit of SWOL arises from its ability to **mitigate representation degradation in internal layers**. In contrast, CoT-VLA’s performance drop likely stems from applying a strong world modeling in pixel space constraint across all layers during fine-tuning, which disrupts the pretrained representations and harms generalization.
>
> | Method        | Task 1 | Task 2 | Task 3 | Task 4 | Task 5 | Avg Len |
> |---------------|--------|--------|--------|--------|--------|---------|
> | $\pi_0$       | 0.914  | 0.760  | 0.626  | 0.494  | 0.380  |  3.17   |
> | DWM           | 0.918	 | 0.765  | 0.626  | 0.504	| 0.412	 |  3.23   |
> | CoT-VLA       | 0.880	 | 0.725  | 0.565  | 0.430	| 0.341	 |  2.94   |
> | SWOL (ours)   | **0.940**  | **0.806**  | **0.666**  | **0.539**  | **0.432**  | **3.38**    |

---

> ### Author Response · Authors · 2025-11-21
>
> >**W3, Q2:** The mid-level representations seems to also vary substantially in quality both within the same architecture (e.g. pi_0's 8th layer does very poorly on semantic classification) and among VLAs (Very noisy for openVLA while very clean for pi_0-Fast). Admitting that the observation is general and empirical, this is still not discussed to a sufficient extent, and the solution does not seem to directly account for this.
>
> **TL;DR: We clarify the phenomenon mentioned above, further justify our choice of mid-layer representations as the target, and propose a simple yet effective solution to mitigate the risks posed by noisy or suboptimally generalized hidden representations.**
>
> Thank you for your valuable comments. We acknowledge that representations can indeed vary across VLAs. However, our analysis reveals two key observations that support our design choice:
>
> 1. **Stronger Overall Generalization in Mid-Layers.** While certain mid-layers (such as layer 8 in $\pi_0$) exhibit weaker semantic generalization, they still demonstrate stronger dynamics generalization compared to deep layers. When jointly considering both semantics and dynamics generalization, mid-layers outperform deep layers overall, ensuring the reliable performance floor when selecting mid-layer as the target for SWOL.
> 2. **Cross-Model Robustness.** The representation of mid-layer demonstrates good generalization across diverse VLA architectures. This architectural consistency significantly reduces the need for hyperparameter tuning when deploying SWOL on new VLA models, enhancing its practicality and scalability.
>
> Finally, to address the problem in choosing the potential noise or suboptimal generalized mid-layer as the target, we propose a simple yet effective solution: using the average of representations from multiple mid-layers (e.g., layers 5–9) as the regression target. This ensemble approach mitigates the risk of anchoring to a poorly generalizing layer. We also conduct the experiments to empirically validate the effectiveness of this solution, as shown below.
>
> | Method      | Task 1 | Task 2 | Task 3 | Task 4 | Task 1 | Avg Len |
> |-------------|--------|--------|--------|--------|--------|----------|
> | $\pi_0$      | 0.914  | 0.760  | 0.626  | 0.494  | 0.380  | 3.17     |
> | SWOL (layer 9) | **0.940**  | **0.806**  | **0.666**  | **0.539**  | 0.432  | **3.38**    |
> | SWOL (mean layer 5-9)   | 0.928  | 0.798  | 0.653  | 0.534  | **0.437**  | 3.35     |
>
> >**W4**: The presentation of the partitioning of the hidden features into perceptual and action parts is not clearly presented. It is unclear to the reader why such a partitioning is taken as a postulate.
>
> We apologize for the lack of clarity regarding the feature partition. Following established conventions in prior work [3,4], VLAs are typically structured into two functional components: a perception module that processes visual and language inputs, and a decision-making (or action) module that generates actions. Crucially, this architectural separation is reflected in the token layout: perceptual tokens (e.g., image patches and language embeddings) and action tokens occupy distinct, fixed positions in the input sequence. This structure allows us to cleanly partition the hidden representation at any layer $l$ as $Z^l=\text{cat}(Z^l_P, Z^l_A)$. where $Z^l_P$ and $Z^l_A$ correspond to the perceptual and action tokens, respectively.
>
> >**W5, W6**: Along these lines, the update rules (lines 112-124) are hard to decipher both because of the above point, but also due to the cumbersome notation. I did not find Figure 1 very useful, especially considering the area/information ratio.
>
> Thank you for your constructive feedback. We have revised the **Section 2.1** to more clearly describe the forward pass of VLAs and explicitly articulate how the hidden representations are partitioned into perceptual and action components. We have also redesigned **Figure 1** to better illustrate the token-level separation between perception and action pathways, improve visual clarity, and increase the information density.

---

> ### Author Response · Authors · 2025-11-21
>
> >**W7, Q3**: In table 1 the best per data fraction and per task score should be in bold as it is fatiguing on the eyes to decipher such a big block of numbers. This is done for the average length but I suspect practitioners care more about success rate.
>
> Thank you for this helpful suggestion. We have bolded the best success rate for each task and data fraction in the tables to improve readability. each evaluation trial requires a policy to **complete a sequence of 5 subtasks in order**, and this process is repeated over 1000 independent trials. The task success rate (reported in columns labeled “Task i”) reflects the percentage of trials in which the policy successfully completed the i-th subtask given that all preceding subtasks were also completed. Additionally, **average task length (Avg Len)**, defined as the average number of consecutive subtasks completed per trial across all 1,000 evaluations, **is the primary performance metric in the CALVIN benchmark**. This metric is directly derived from the individual task success rates and can be computed as the sum of the success rates of 5 tasks.
>
> >**W8**: The paper fails to report a reproducibility statement as well as a statement on the use of LLMs which I believe are required.
>
> The use of LLM has been included in **Appendix A**. Additionally, we have added a reproducibility statement in **Appendix B** of the revised paper. We commit to open-sourcing training data and code to support reproducibility.
>
> ----
> ### References
> [1] General Agents Contain World Models. ICML’25.
>
> [2] CoT-VLA: Visual Chain-of-Thought Reasoning for Vision-Language-Action Models. CVPR’25.
>
> [3] Vla-Adapter: An Effective Paradigm for Tiny-scale Vision-Language-Action
> Model. AAAI'25.
>
> [4] VLA-0: Building State-of-the-Art VLAs with Zero Modification.

---

> > ### Comment · Reviewer_by5U · 2025-11-27
> > **Follow-up questions to authors**
> >
> > I would like to thank the authors for the clarifications provided. In whole they have allowed me to better understand details of the work that were opaque.
> >
> > I have a couple of follow up questions:
> >
> > 1) First regarding the new world modeling baselines. The authors mention:
> >
> > "Direct World Modeling (DWM). We apply the world modeling loss using the first-layer representation as input, ensuring that the dynamics loss does not influence or regularize the degraded hidden-layer representations."
> >
> > Why is this approach limited to the first layer in such a way. Can the objective not be to improve the degraded layers directly ? In other words, is this baseline intentionally restrictive ?
> >
> > 2) I will push the question regarding the choice of mid-layers one step further. The authors propose an ensemble approach which is appreciated. Why 5-9 not 4-8 or 6-10 or any other combination. In other words what is a mid-layer to start with, where and how do you draw the limit?
> >
> > 3) As a general note I appreciate the clarification regarding the world modeling capability of layers and predictability of future states being necessary for general agents. I think the authors should lead with this argument in the presentation of the approach in the manuscript for it to resonate with a wider audience. Overall, the presentation of arguments in the revised manuscript still lacks complete crispness.

---

> > > ### Author Response · Authors · 2025-11-30
> > >
> > > We sincerely thank the reviewer for their thoughtful feedback and constructive suggestions. We would like to clarify these further questions:
> > >
> > > > **Q1**: Regarding the new world modeling baselines. Why is this approach limited to the first layer in such a way? Can the objective not be to improve the degraded layers directly?
> > >
> > > **TL;DR: We clarify the motivation behind the DWM design and introduce an additional baseline that applies the world-modeling loss directly to the degraded layers.**
> > >
> > > We appreciate the reviewer's valuable comment and would like to provide clarification. DWM is deliberately designed to test whether adding any "world modeling loss alone" suffices to improve performance. By applying this loss exclusively to the first-layer features while keeping all other settings unchanged, we prevent gradient updates from reaching the deeper layers.
> > >
> > > Here, we add another baseline (DWMD) below, where the input and target layer of world modeling loss are both the last layer (degraded hidden layer). Although these models also incorporate the world modeling loss, the degradation problem in the input layer cannot be addressed because the target layer itself is degraded. The results are shown below. Moreover, we find that **DWMD leads to highly unstable training, with frequent loss spikes.**
> > >
> > > | Method  | Input Layer | Target Layer   | Task 1 | Task 2 | Task 3 | Task 4 | Task 5 | Avg Len |
> > > |-|-|-|-|-|-|-|-|-|
> > > | $\pi_0$ | -| - | 0.914  | 0.760  | 0.626| 0.494| 0.380| 3.17|
> > > | DWM  |1 | 1 | 0.918  | 0.765  | 0.626  | 0.504  | 0.412|  3.23  |
> > > | DWMD |17  | 17| 0.924 | 0.791 |0.626 |0.516 |0.405 |3.26 |
> > > | SWOL | 17 | 9 | **0.940**  | **0.806** | **0.666**| **0.539**| **0.432** | **3.38** |
> > >
> > > > **Q2**: On the mid-layer ensemble (5–9) approach, definition and choice of "mid-layers".
> > >
> > > **TL;DR: We clarify the definition of “mid-layers”, justify the choice of the layer ensemble (5–9), add a new ablation study on layer 6-10 ensemble, and provide a practical, flexible protocol for selecting target layers in new VLA models.**
> > >
> > > Thank you for your valuable questions. Previous studies have shown that the representations of large models can be roughly partitioned into early, middle, and late stages [1, 2, 3]. These works explicitly refer to the middle block as “mid-layer” representations, which are typically more stable than those in the early or late layers. However, since different models exhibit varying behaviors, “mid-layer” is generally used descriptively rather than as a strictly defined boundary. In our paper, we adopt this terminology. We define mid-layers as the central layers of the model or the layers near the central region, which have better generalization than those in the deep layers.
> > >
> > > Crucially, SWOL’s protocol for selecting the target layer is **not based on absolute layer indices**, but **on representation quality**. In practice, we find that VLA models consistently exhibit such high-quality representations in their mid-layers. To keep the method simple and easy to apply across different models, **we use the exact central layer of each model as the practical default target**. For example, $\pi_0$ and $\pi_0$-Fast each have 18 layers, so we target layer 9. OpenVLA-OFT has 32 layers, and we accordingly select layer 16.
> > >
> > > The additional experiments over layers 5–9 were conducted in response to Reviewer 5mNv’s. We extend our analysis to layers 6–10, as shown in the table below. SWOL’s performance remains stable, confirming that its effectiveness is not sensitive to minor shifts in the selected layer interval.
> > >
> > > | Method| Task 1 | Task 2 | Task 3 | Task 4 | Task 5 | Avg Len |
> > > |-|-|-|-|-|-|-|
> > > | $\pi_0$ |0.914|0.760|0.626|0.494|0.380|3.17|
> > > | SWOL|**0.940**|**0.806**|**0.666**|**0.539**|0.432|**3.38**|
> > > | SWOL (layer 5-9) |0.928|0.798|0.653|0.534|**0.437**|3.35|
> > > | SWOL (layer 6-10) |0.931|0.805|0.659|0.536|0.425|3.36|
> > >
> > > In summary, SWOL is robust to the choice of target layer. For practitioners applying our method to new architectures, we recommend one of the following strategies:
> > > 1. Use the exact central layer (e.g., layer $L//2$ for an $L$-layer model) as a simple default.
> > > 2. Average features across the central layer and its immediate neighbors for potentially improved stability.
> > > 3. Perform a representation analysis to empirically identify the layer with the strong generalization in dynamics and semantics, and use that as the target.
> > >
> > > > **Q3**: Presentation & Motivation.
> > >
> > > Thank you for your valuable comment. We agree with your point that the manuscript should more prominently highlight the importance of world modeling capability. We have revised the **Section 3.1 (Key Insight)** in the paper to emphasize this perspective more clearly and position it as a central motivation for our approach.
> > >
> > > ---
> > > ### References
> > > [1] Do Vision Transformers See Like Convolutional Neural Networks? NeurIPS'21.
> > >
> > > [2] Transformer Layers as Painters. AAAI'24.
> > >
> > > [3] Layer by Layer: Uncovering Hidden Representations in Language Models. ICML'25.

---

### Official Review · Reviewer_5mNv · 2025-10-26

**Soundness:** 4
**Presentation:** 3
**Contribution:** 3
**Rating:** 6
**Confidence:** 3

**Summary:**

The paper “On the Representation Degradation in Vision-Language-Action Models” presents an in-depth study of how internal representations evolve across layers in Vision-Language-Action (VLA) models. The authors uncover a consistent and concerning trend: representation degradation, where deeper layers,  responsible for generating actions,  lose generalization capacity to both semantic and dynamic aspects of the environment.
To address this, the paper proposes SWOL (Hidden Space World Modeling), a lightweight auxiliary objective that aligns degraded deep-layer features with mid-layer representations from future observations. SWOL effectively introduces a self-supervised “world modeling” signal in hidden space without architectural modifications or inference overhead.
Extensive experiments on CALVIN (simulation) and real-world robotic tasks (Aloha/ARX5 platform) show that SWOL improves policy generalization, particularly in low-data settings, enhancing both semantic grounding and dynamic awareness of VLAs.

**Strengths:**

- The discovery of representation degradation fills an important analytical gap. By decomposing the forward pass and probing layer-wise generalization, the authors provide valuable interpretability into how semantic and dynamic information dissipates through depth.

- SWOL stands out for its conceptual clarity: it uses mid-layer representations from future observations to “rejuvenate” degraded deep-layer embeddings. This plug-and-play auxiliary loss is elegant, general, and requires no changes to model architecture or inference.

- The authors test SWOL across multiple VLA architectures (π₀, π₀-fast, OpenVLA-OFT), different data regimes (1%, 10%, 100%), and both simulated and real-world environments. The consistent improvements across setups strengthen the empirical claim.

- Unlike many representation analysis papers confined to simulation, this work extends experiments to real manipulation tasks like folding a towel, plugging in a cable, and cleaning a table. These practical gains significantly bolster the contribution’s impact.

- The study includes careful ablations on loss weight, target layer, and predictor architecture, along with visualization of improved deep-layer representation quality. This rigor demonstrates strong experimental maturity.

**Weaknesses:**

- The connection between SWOL and formal notions of world modeling remains intuitive rather than mathematically grounded. The paper could benefit from a clearer theoretical explanation of why aligning to future mid-layer features enhances generalization beyond empirical observation.

- All experiments focus on imitation learning scenarios. While the method should, in principle, generalize to reinforcement learning or planning-based agents, this is not tested or discussed in detail.

- Although the authors claim “no inference overhead,” SWOL roughly increases training GPU hours by 25–30%. This discrepancy could be better contextualized.

- Similar auxiliary consistency losses (e.g., temporal or cross-view prediction) have been explored in visual representation learning. The paper’s differentiation from these paradigms could be more explicit.

- Aligning deep features to mid-layer targets risks encouraging representational homogeneity. While empirical results show gains, a discussion on possible over-regularization effects is missing.

**Questions:**

- How does SWOL compare against standard temporal consistency or contrastive predictive coding baselines in representation learning?

- Does the improvement persist if the target mid-layer is randomly sampled instead of fixed (e.g., layers 5–9)?

- Could SWOL interfere with the diversity of action representations by enforcing excessive alignment across time?

- Could the authors provide qualitative visualization (e.g., t-SNE) of how representation structure changes before vs. after SWOL?

---

> ### Author Response · Authors · 2025-11-21
>
> We sincerely appreciate the reviewer’s constructive feedback and would like to offer further clarification in response.
>
> >**W1**:The connection between SWOL and formal notions of world modeling remains intuitive rather than mathematically grounded.
>
> **TL;DR: We answer this by disentangling two components of SWOL’s design: **(i) its theoretical reliance on predicting future states** as targets, and **(ii) its use of mid-layer features as predictive representations**.**
>
> Thank you for your valuable question. We clarify that SWOL’s effectiveness stems from two distinct design choices, which we analyze separately below.
>
> 1. **The role of predicting future features.** Previous work [1] has established that any agent satisfying a regret bound over composite goals of depth $n > 1$ **must** implicitly encode an approximation transition $\hat{P}(s' \mid s, a)$ of the true environment transition function $P(s' \mid s, a)$, with error $\bigl| \hat{P}(s' \mid s, a) - P(s' \mid s, a) \bigr| = \mathcal{O}\left(\frac{\delta}{\sqrt{n}}\right) + \mathcal{O}\left(\frac{1}{n}\right),$ where $\delta$ is the agent’s maximum regret and $n$ is the goal depth. Crucially, the transition function $P(s' \mid s, a)$ is defined precisely as the conditional distribution over future states $s'$ given current state $s$ and action $a$. Therefore, any mechanism that enforces predictability of future states from current state-action pairs is not a heuristic but a **necessary condition** for general agents. Motivated by this theorem, SWOL’s auxiliary loss, which minimizes the distance between predicted and actual future representations, explicitly enforces this condition, thereby ensuring that the agent’s policy satisfies the preconditions under which a world model is provably recoverable.
> 2. **The role of predicting mid-layer features.** In this paper, the core issue we aim to address is the empirical phenomenon of representation degradation in VLAs. The reason we use mid-layer as the target is that we observe that **middle-layer representations across different VLAs are consistently robust**. **To reduce tuning overhead when applying SWOL to various VLAs**, we default to using a mid-layer as the target for practicality and stability.
>
> >**W2**: All experiments focus on imitation learning scenarios. While the method should, in principle, generalize to reinforcement learning or planning-based agents, this is not tested or discussed in detail.
>
> **TL;DR: We conduct RL experiments for VLA models and demonstrate both the applicability and the superior performance of SWOL.**
>
> Thank you for valuable comment. While our main experiments focus on imitation learning, our method is, in principle, applicable to RL and planning-based agents as well. To substantiate this claim, we include a new RL experiment on the Libero Spatial benchmark [2], a widely used benchmark for VLA-RL.
>
> Specifically, we begin with a 5-shot SFT $\pi_0$ model and perform RL via Reinflow [3] with LoRA. During policy updates, we incorporate a hidden-space world modeling loss analogous to SWOL. Our results show that SWOL achieves a best success rate of **92.2%**, compared to **85.9%** for the baseline (w.o. SWOL), demonstrating that our approach effectively generalizes to RL settings. Full training curves and implementation details are provided in **Appendix E.8** of the revised paper.
>
> >**W3**: Although the authors claim “no inference overhead,” SWOL roughly increases training GPU hours by 25–30%. This discrepancy could be better contextualized.
>
> We apologize for the omission of clarification of the increased training computation cost. The increased computational cost primarily stems from the additional forward pass required to compute mid-layer features of future states. However, since SWOL does not modify the VLA’s architecture or inference procedure, it introduces no extra computation at inference time. We have provided a more detailed explanation of this in the revised version of our paper.

---

> ### Author Response · Authors · 2025-11-21
>
> > **W4**: The paper’s differentiation from similar auxiliary consistency losses in visual representation learning could be more explicit.
>
> **TL;DR:  We clarify the differences from the scope, objective, and method perspectives.**
>
> Thank you for your valuable comment. Compared to auxiliary consistency losses in visual representation learning (e.g., temporal or cross-view prediction), SWOL differs primarily in several key aspects:
> 1. **Scope**: SWOL is designed for **decision-making**, particularly VLA models, whereas visual representation learning focuses on perception-centric tasks and vision models.
>
> 2. **Objective**: SWOL aims to address the **representation degradation** in VLA models, which is also a new empirical finding in our work. In contrast, visual representation learning typically seeks to learn invariant or predictive visual features for recognition, detection, or reconstruction.
>
> 3. **Method**: SWOL introduces a hidden-space world modeling loss that enforces consistency between predicted and actual mid-layer representations of future states, a mechanism not commonly explored in visual representation learning.
>
>     These distinctions highlight SWOL’s unique role in enhancing structured reasoning and planning in embodied agents, rather than just improving perceptual representations. We have added these discussions to our revised paper.
>
> >**W5**: Aligning deep features to mid-layer targets risks encouraging representational homogeneity. While empirical results show gains, a discussion on possible over-regularization effects is missing.
>
> In principle, adding an auxiliary loss like SWOL may introduce possible over-regularization effects. However, by appropriately tuning the weight of this auxiliary loss, we can effectively balance representation regularization and action prediction, ensuring that the auxiliary objective enhances rather than compromises the policy performance. Our ablation studies in **Appendix E.3** also show that SWOL’s performance is not sensitive to the choice of this hyperparameter.
>
> >**Q1**: How does SWOL compare against standard temporal consistency or contrastive predictive coding baselines in representation learning?
>
> **TL;DR: We add three baselines, Direct World Modeling (DWM), Direct World Modeling-Deep (DWMD), and CoT-VLA [4], and show that SWOL consistently performs better than these baselines.**
>
> Thank you for your insightful feedback. We add three baselines:
> 1. **Direct World Modeling (DWM).** We apply the world modeling loss using the first-layer representation as input, ensuring that the dynamics loss **does not influence or regularize the degraded hidden-layer representations.**
> 2. **Direct World Modeling-Deep (DWMD).** Similar to DWM, where the input and target layer of world modeling loss are both the last layer (degraded hidden layer).
> 3. **CoT-VLA [1]**: Following the original formulation, we reimplement Cot-VLA on top of the $\pi_0$ architecture to ensure a fair comparison. We extend the input sequence by appending future image tokens after the current image and language tokens, and introduce an additional image decoder head trained with a world modeling loss that **predicts these future frames directly in pixel space**.
>
> | Method| Task 1 | Task 2 | Task 3 | Task 4 | Task 5 | Avg Len |
> |-|-|-|-|-|-|-|
> | $\pi_0$  | 0.914  | 0.760  | 0.626  | 0.494| 0.380| 3.17|
> | DWM  | 0.918| 0.765  | 0.626  | 0.504| 0.412|  3.23|
> | DWMD | 0.924 | 0.791 |0.626 |0.516 |0.405 |3.26 |
> | CoT-VLA | 0.880| 0.725  | 0.565  | 0.430| 0.341 |  2.94|
> | SWOL (ours) | **0.940**  | **0.806**  | **0.666**  | **0.539**| **0.432**| **3.38** |
>
> The results demonstrate that SWOL consistently outperforms all other baselines. The limited improvement from DWM and DWMD suggests that merely adding a predictive loss is insufficient; the key benefit of SWOL arises from its ability to mitigate representation degradation in internal layers. In contrast, CoT-VLA’s performance drop likely stems from applying a strong pixel-based world modeling constraint across all layers during fine-tuning, which disrupts the pretrained representations and harms generalization.
>
> >**Q2**: Does the improvement persist if the target mid-layer is randomly sampled instead of fixed (e.g., layers 5–9)?
>
> Thank you for the insightful question. We add two additional implementations to answer this question. (1) Randomly choose a layer from 5-9 layers as the target. (2) Use the mean representation of 5-9 layers as the target. The results are shown below, showing that these two approaches consistently improve the policy performance of the VLA model.
>
> | Method | Task 1| Task 2 | Task 3 | Task 4 | Task 1 | Avg Len |
> |-|-|--|-|-|-|-|
> | $\pi_0$ | 0.914 | 0.760|0.626  | 0.494|0.380  | 3.17 |
> | SWOL (layer 9) | **0.940**| **0.806** | **0.666** | **0.539**  | 0.432| **3.38**|
> | SWOL (random layer 5-9)| 0.927| 0.792| 0.633 | 0.517| 0.397| 3.27|
> | SWOL (mean layer 5-9)| 0.928 | 0.798  | 0.653  | 0.534  | **0.437**  | 3.35|

---

> ### Author Response · Authors · 2025-11-21
>
> >**Q3**: Could SWOL interfere with the diversity of action representations by enforcing excessive alignment across time?
>
> **SWOL does not interfere with the diversity of action representations. Oppositely, SWOL improves the diversity.** To validate this, we visualize and quantify the diversity of representations across different timesteps within the same trajectory. As shown in **Figure 14 in Appendix E.9**., the representations of a single episode become more spatially dispersed after applying SWOL, indicating improved diversity in action representations. This observation is further validated by quantitative metrics: the mean centroid distance increases **from 23.63 to 46.46**, and the convex hull area grows from **219.73 to 385.23**, confirming consistent diversity improvement of SWOL. The more detailed experimental setup, see **Appendix E.9**.
>
>
> >**Q4**: Could the authors provide qualitative visualization (e.g., t-SNE) of how representation structure changes before vs. after SWOL?
>
> We add the qualitative t-SNE visualization to illustrate how representations evolve with SWOL. We randomly select several image-text pairs of different tasks and generate the hidden representation for t-SNE visualization. As show in **Figure 15 in Appendix E.9** , the representation before SWOL is mixed together among third-person camera images, gripper camera images, and text, whereas clear separation emerges after SWOL. **This marked improvement in representation structure demonstrates that SWOL enhances multimodal understanding of VLAs.**
>
> ----
> ### References
> [1] General Agents Contain World Models. ICML’25.
>
> [2] Libero: Benchmarking Knowledge Transfer for Lifelong Robot Learning. NeurIPS'23.
>
> [3] Reinflow: Fine-tuning Flow Matching Policy with Online Reinforcement Learning. NeurIPS'25.
>
> [4] CoT-VLA: Visual Chain-of-Thought Reasoning for Vision-Language-Action Models. CVPR’25.

---

### Official Review · Reviewer_Dbfu · 2025-10-30

**Soundness:** 3
**Presentation:** 3
**Contribution:** 3
**Rating:** 6
**Confidence:** 4

**Summary:**

This paper discovers that semantic and dynamic representations degrade with network depth in VLA models, reducing generalization. To verify this, the authors use task intent classification for semantic generalization and inverse dynamics prediction for dynamic generalization. They propose SWOL, a simple yet effective approach, which makes deep-layer features predict mid-layer representations from future observations, helping to recover lost information and mitigate degradation.​ The authors conduct extensive experiments on the CALVIN simulation benchmark and ARX5 robotic platform, testing $\pi_0$, $\pi_0$-fast, and OpenVLA-OFT and showing SWOL's superiority. Ablation studies analyze key design choices, and further analysis of semantic and dynamic prediction results confirms SWOL's effectiveness. The paper's main contributions are identifying representation degradation as a key issue in VLA models and proposing the plug-and-play SWOL method to address it.

**Strengths:**

1. This paper rigorously identifies the representation degradation phenomenon in fine-tuned VLA models, which is an overlooked issue in prior VLA representation research.

2. The authors design two well-targeted evaluation protocols: semantic intent classification and inverse dynamics regression, systematically measure the distribution of semantic and dynamic information across layers, clearly revealing that mid-layers maintain strong representational quality while deeper layers suffer from significant degradation.

3. The proposed SWOL method is innovative in its design. By performing future mid-layer representation prediction in the hidden space, it avoids the computational inefficiency and appearance sensitivity of raw visual-space future prediction, while forcing the model to re-learn degraded representations. It's plug-and-play, enabling seamless integration with various existing VLA models.

**Weaknesses:**

1. The paper lacks quantitative comparative experiments with model-based methods discussed in the Related Works section.

2. Typo: The fourth legend in Figure 5 should be "Dyn. w. SWOL"

**Questions:**

1. The paper posits that direct visual-space modeling is highly susceptible to appearance variations. However, it lacks robust quantitative evidence demonstrating the superiority of SWOL. Are there comparative experiments showcasing SWOL's performance edge over conventional visual future prediction methods? Specifically, tests should involve varying critical factors such as lighting conditions, object textures, and background clutter to comprehensively validate the robustness claims.

2. Given that SWOL adds an auxiliary loss during training, does it introduce any unintended side effects, such as overfitting to mid-layer representations or compromising the original action generation capability of VLA models? If so, how are these trade-offs managed?

---

> ### Author Response · Authors · 2025-11-21
>
> We sincerely appreciate the reviewer’s constructive feedback and would like to offer further clarification in response.
>
> > **W1**: The paper lacks quantitative comparative experiments with model-based methods discussed in the Related Works section.
>
> **TL;DR: We add three model-based baselines, Direct World Modeling (DWM), Direct World Modeling-Deep (DWMD), and CoT-VLA [1], and show that SWOL consistently performs better than these baselines.**
>
> Thank you for your insightful feedback. We add three baselines:
> 1. **Direct World Modeling (DWM).** We apply the world modeling loss using the first-layer representation as input, ensuring that the dynamics loss **does not influence or regularize the degraded hidden-layer representations.**
> 2. **Direct World Modeling-Deep (DWMD).** Similar to DWM, where the input and target layer of world modeling loss are both the last layer (degraded hidden layer).
> 3. **CoT-VLA [1]**: Following the original formulation, we reimplement Cot-VLA on top of the $\pi_0$ architecture to ensure a fair comparison. We extend the input sequence by appending future image tokens after the current image and language tokens, and introduce an additional image decoder head trained with a world modeling loss that **predicts these future frames directly in pixel space**.
>
> | Method| Task 1 | Task 2 | Task 3 | Task 4 | Task 5 | Avg Len |
> |-|-|-|-|-|-|-|
> | $\pi_0$  | 0.914  | 0.760  | 0.626  | 0.494  | 0.380  |  3.17   |
> | DWM  | 0.918| 0.765  | 0.626  | 0.504	| 0.412|  3.23   |
> | DWMD  | 0.924 | 0.791 |0.626 |0.516 |0.405 |3.26 |
> | CoT-VLA  | 0.880	 | 0.725  | 0.565  | 0.430	| 0.341	 |  2.94   |
> | SWOL (ours)   | **0.940**  | **0.806**  | **0.666**  | **0.539**  | **0.432**  | **3.38**    |
>
> The results demonstrate that SWOL consistently outperforms all other baselines. The limited improvement from DWM and DWMD suggests that merely adding a predictive loss is insufficient; the key benefit of SWOL arises from its ability to mitigate representation degradation in internal layers. In contrast, CoT-VLA’s performance drop likely stems from applying a strong pixel-based world modeling constraint across all layers during fine-tuning, which disrupts the pretrained representations and harms generalization.
>
> >**W2**: Typo: The fourth legend in Figure 5 should be "Dyn. w. SWOL".
>
> Thank you for pointing out this typo. We have corrected the legend in this figure to "Dyn. w. SWOL" in the revised version.
>
> >**Q1**:  The paper posits that direct visual-space modeling is highly susceptible to appearance variations. However, it lacks robust quantitative evidence demonstrating the superiority of SWOL. Are there comparative experiments showcasing SWOL's performance edge over conventional visual future prediction methods? Specifically, tests should involve varying critical factors such as lighting conditions, object textures, and background clutter to comprehensively validate the robustness claims.
>
> **TL;DR: We clarify our statements regarding direct visual-space modeling and present additional real-world experiments under more visually challenging conditions to demonstrate the visual robustness of SWOL.**
>
> We sincerely apologize for the ambiguity in our original wording that may have misled the reviewer. In lines 464-468, our claim **was not that visual-space modeling causes sensitivity to appearance changes during policy inference,** but rather that **directly predicting future frames in pixel space is sensitive to appearance variations during training**. This sensitivity can increase training difficulty, raise computational costs, and ultimately degrade policy performance. This issue is empirically supported by our simulation experiments, where CoT-VLA performs worse than SWOL and other baselines.
>
> To further validate the visual robustness of SWOL, we conduct more challenging real-robot experiments with increased background clutter. These experiments are performed on the Pick\&Place task. We design three more challenging scenes, considering background clutter and lighting conditions: (i) Black table background. (ii) Green table background. (iii) Color-changing lights. The results are shown in the Table below. Performance follows a ranking of $\pi_0$-SWOL > $\pi_0$-DWMD > $\pi_0$ > $\pi_0$-DWM > $\pi_0$-CoT-VLA, with SWOL exhibiting the strongest robustness to environmental disturbances. For detailed setups, see **Appendix E.6**.
>
> | Modeling Method | Black Background | Green Background | Changing Light | Avg. Success Rate |
> |--|----|----|----|----|
> | $\pi_0$  | 19/30   | 19/30  | **18/30**      | 62.2  |
> | DWM  | 18/30 | 19/30    | 17/30    | 60.0 |
> | DWMD  | 20/30  | 20/30   | 17/30    | 63.3    |
> | CoT-VLA  | 16/30     | 17/30   | 15/30  | 53.3   |
> | SWOL (ours)    | **22/30** | **21/30** | 18/30 | **67.8**  |

---

> ### Author Response · Authors · 2025-11-21
>
> >**Q2**: Given that SWOL adds an auxiliary loss during training, does it introduce any unintended side effects, such as overfitting to mid-layer representations or compromising the original action generation capability of VLA models? If so, how are these trade-offs managed?
>
> In principle, adding an auxiliary loss like SWOL may introduce unintended side effects, such as overfitting to mid-layer representations or interfering with the original action generation capability of VLA models. However, by appropriately tuning the weight of this auxiliary loss, we can effectively balance representation regularization and action prediction, ensuring that the auxiliary objective enhances rather than compromises the policy performance. Our ablation studies in **Appendix E.3** also show that SWOL’s performance is not sensitive to the choice of this hyperparameter.
>
> ----
> ### References
> [1] CoT-VLA: Visual Chain-of-Thought Reasoning for Vision-Language-Action Models. CVPR’25.

---

### Official Review · Reviewer_iHK4 · 2025-10-31

**Soundness:** 3
**Presentation:** 3
**Contribution:** 2
**Rating:** 4
**Confidence:** 4

**Summary:**

The paper finds a layer-wise representation degradation phenomenon in fine-tuned Vision-Language-Action (VLA) models, losing task semantics and dynamics information in the deep layers. Then, this paper proposes SWOL (Hidden Space World Modeling), training an alignment between deep-layer features to mid-layer features of the next observation with a simple MLP predictor. SWOL has no additional inference cost, and yields consistent gains on CALVIN and in real-robot manipulation tasks.

**Strengths:**

- Generalization of VLA models is an important research problem.
- The paper empirically observes representation degradation and low-complexity intervention that efficiently solves the problem with improvements in simulated and real world settings.
- Results span multiple VLA backbones, low-data regimes, long-horizon tasks, and real-robot experiments.
- The method has no inference overhead, making it attractive and easy for applied use.

**Weaknesses:**

- Unclear necessity of correcting representation degradation: Many VLA architectures condition action decoding on all intermediate features of VLM. If the action expert can access earlier semantically rich features, it is not obvious why degradation in some deep layers must be corrected since there could be semantics agnostic behavior in the deep layers such as precise refinement of actions. Performance gain could be purely from integration of dynamics in the world model objective.

- Insufficient comparison to prior world-modeling baselines:  From prior works [1, 2], performance gains from predictive auxiliary objectives like implicit or explicit next-state prediction is not that surprising. This work does not convincingly separate SWOL’s benefits from these approaches. Direct empirical comparisons to at least a couple of simple representative baselines are necessary.

[1] Zhao, Qingqing et al., CoT-VLA: Visual Chain-of-Thought Reasoning for Vision-Language-Action Models.

[2] Zheng, Ruijie et al., FLARE: Robot Learning with Implicit World Modeling.

**Questions:**

- How well does representation semantics and dynamics experiments perform with pretrained VLM weights? Discrete action tokens tend to less disturb LLM’s representation space but also appear in discrete action models, so representation degradation could be related to biased behavior from the pretrained weight.
- In ablation, using the first layer as alignment target shows best performance. Using the first layer as target is nearly equivalent to CoT-VLA, except output tokens are from the perception tokens and targets are from layer 1 (which is close to input representation), questioning the need of mid-layer alignment where task semantics and dynamics are theoretically upper bounded by that of input representations.
- The result of target layer 5 in Table 3 appears missing.

---

> ### Author Response · Authors · 2025-11-21
>
> We sincerely appreciate the reviewer’s constructive feedback and would like to offer further clarification in response.
>
> > **W1**: Unclear necessity of correcting representation degradation. If the action expert can access earlier semantically rich features, it is not obvious why degradation in some deep layers must be corrected.
>
> **TL;DR: We empirically show that representation degradation harms the performance of VLAs, even in models with access to early-layer features, making correction necessary.**
>
> We appreciate the reviewer's valuable comment and would like to provide clarification. In our work, the VLA models we employ (e.g., $\pi_0$ and $\pi_0$-Fast) condition their action predictions on features from all intermediate layers. **We empirically find that the generalizability in deep layers significantly affects the action generation in such models, making it necessary to address the generalization representation degradation.** Specifically, we conduct a layer-wise zero-ablation study following a methodology similar to that in [1, 2]. We replace the representation at the i-th hidden layer with zero values during inference, effectively simulating a non-informative or poorly generalizable representation. As shown in **Figure 10 in Appendix E.4**, this replacement leads to a substantial drop in task success rates, with deeper layers showing the greatest sensitivity. For instance, replacing the representation in the final layer alone causes a performance drop of more than 20%. These results strongly indicate that hidden representations in VLAs are crucial for shaping robust policy behavior. Thus, preventing representation degradation throughout the network is essential.
>
> > **W1**: Performance gain could be purely from the integration of dynamics in the world model objective.
>
> **TL;DR: The performance gain arises from both mitigating the representation degradation and the integration of world modeling.**
>
> Thank you for the thoughtful question. To better isolate the source of the performance gains in our method, we introduce two additional baselines, referred to as Direct World Modeling (DWM) and Direct World Modeling-Deep (DWMD). In DWM, we apply the world modeling loss using the first-layer representation as input, ensuring that the dynamics loss does not propagate to or regularize the degraded hidden-layer representations. In DWMD, the input and target layer of world modeling loss are both the last layer (degraded hidden layer). As shown in the results below, these approaches offer an improvement over BC, but still fall short of SWOL by a clear margin. This reveals that the superior performance of SWOL is not solely due to the world modeling objective itself, but also stems from mitigating the representation degradation.
>
> | Method    | Input Layer | Target Layer   | Task 1 | Task 2 | Task 3 | Task 4 | Task 5 | Avg Len |
> |-|-|-|-|-|-|-|-|-|
> | $\pi_0$| -| - | 0.914  | 0.760  | 0.626  | 0.494  | 0.380| 3.17|
> | DWM  |1 | 1 | 0.918  | 0.765  | 0.626  | 0.504  | 0.412|  3.23  |
> | DWMD |17  | 17| 0.924 | 0.791 |0.626 |0.516 |0.405 |3.26 |
> | SWOL | 17 | 9 | **0.940**  | **0.806**  | **0.666**  | **0.539**| **0.432** | **3.38** |

---

> ### Author Response · Authors · 2025-11-21
>
> > **W2**: Insufficient comparison to prior world-modeling baselines.
>
> **TL;DR: We add CoT-VLA [3] as a world-modeling baseline and show that SWOL consistently performs better than these methods.**
>
> Thank you for your valuable suggestions. Since the two baselines mentioned above are not open-sourced, we reimplement CoT-VLA on top of the $\pi_0$ architecture to ensure a fair comparison. Specifically, we extend the input sequence by appending future image tokens after the current image and language tokens, and introduce an additional image decoder head trained with a world modeling loss that **predicts these future frames directly in pixel space**. We carefully validated the correctness of our implementation and performed simple hyperparameter tuning to optimize CoT-VLA’s performance. The overall results are summarized in the table below.
>
> Our experiments demonstrate that naively applying CoT-VLA to off-the-shelf VLA models can actually degrade generalization, while our method improves the policy performance. The underperformance of CoT-VLA may stem from the fact that its original implementation uses the dynamics loss in pixel space during pretraining, while our version applies it only during fine-tuning. Imposing such a strong inductive bias after pretraining, especially across all layers, may corrupt pretrained representations and harm generalization. In contrast, SWOL selectively regularizes only the degradation features, preserving the model’s original capabilities. This targeted intervention not only improves generalization but also makes SWOL plug-and-play.
>
> | Method        | Task 1 | Task 2 | Task 3 | Task 4 | Task 5 | Avg Len |
> |---------------|--------|--------|--------|--------|--------|---------|
> | $\pi_0$       | 0.914  | 0.760  | 0.626  | 0.494  | 0.380  |  3.17   |
> | CoT-VLA       | 0.880	 | 0.725  | 0.565  | 0.430	| 0.341	 |  2.94   |
> | SWOL (ours)   | **0.940**  | **0.806**  | **0.666**  | **0.539**  | **0.432**  | **3.38**    |
>
> > **Q1**: How well do representation semantics and dynamics experiments perform with pretrained VLM weights?
>
> **TL;DR: We conduct semantics and dynamics experiments on a pretrained VLM model and find that (i) Training on robot data reduces the semantic generalization of VLM. (ii) Training on robot data enhances the dynamics generalization of VLM. (iii) All models exhibit varying degrees of representation degradation, with the degradation being more pronounced after pretraining and finetuning on robot data.**
>
> Thank you for the valuable question. The current VLA training process can be broadly divided into three stages: **Pretrained VLM → Pretrain VLA on Large-Scale, Diverse Robot Data → Finetune VLA on Task-Specific Robot Data**. We analyze the models at each of these stages for $\pi_0$-Fast, a discrete action decoding VLA model, with the results presented in **Figure 12 in Appendix E.7** of the revised paper. Our experimental analysis reveals distinct trends in the representation layers of these models. Based on these findings, we draw the following conclusions:
>
> 1. Pretraining on robot data **reduces** the **semantic generalization** of VLM.
> 2. Pretraining on robot data **enhances** the **dynamics generalization** of VLM.
> 3. All models exhibit varying degrees of **representation degradation**, with the degradation being more pronounced after pretraining and finetuning on robot data. However, the **SWOL** technique significantly alleviates this degradation.
>
> The underlying reasons for the first two conclusions may lie in the fact that, during VLM pretraining, the model has not been exposed to a large volume of robot data. Consequently, when VLA is trained on data with a significantly different distribution (robot data), it disrupts some of the semantic representations within the VLM while simultaneously boosting its dynamics generalization.

---

> ### Author Response · Authors · 2025-11-21
>
> > **Q2**: In ablation, using the first layer as alignment target shows best performance. Using the first layer as target is nearly equivalent to CoT-VLA, questioning the need of mid-layer alignment where task semantics and dynamics are theoretically upper bounded by that of input representations.
>
> **TL;DR: We clarify why the first layer performs well as an alignment target, justify our practical choice of mid-layers as the default, and highlight key architectural differences between SWOL and CoT-VLA.**
>
> We appreciate the reviewer's valuable comment and would like to provide clarification. Firstly, our proposed challenge **representation degradation mainly focuses on the degradation of hidden layers rather than asserting whether early or middle layers are inherently superior.** This framing allows us to select as the alignment target any layer whose representations generalize well. According to the **Figure 3** in the paper, $\pi_0$’s first layer demonstrates the best semantic generalization, which may explain why using the first layer as the alignment target yields the strongest performance in ablation studies.
>
> Moreover, we agree that the first layer retains the richest task semantics and dynamical information, as it has not yet undergone the representational compression (and associated information loss). However, the generalizability of a representation is not solely determined by the amount of information it contains; other factors, such as compression ratio, smoothness, and structural coherence, also play critical roles. Indeed, **we find that only mid-layer representations consistently generalize well across different VLAs**. For instance, in $\pi_0-$Fast, the first-layer representations exhibit poor semantic generalizability. **To reduce tuning overhead when applying SWOL to various VLAs, we default to using a middle layer as the target for practicality and stability.**
>
> Furthermore, even when both SWOL use the first layer as the target, it differs from CoT-VLA significantly in design. (i) CoT-VLA employs world modeling **in pixel space**, while our approach applies self-consistent world modeling in the hidden space. (ii) CoT-VLA directly extends the input sequence by appending future image tokens to the current image and language tokens, which **influences the entire Transformer backbone**. In contrast, SWOL selectively regularizes only the degradation features, leaving the rest of the pretrained representations intact. This targeted intervention preserves the model’s original capabilities and allows SWOL to be truly plug-and-play. Our additional baselines comparison experiments further validate the need of mid-layer alignment.
>
> > Q3: The result of target layer 5 in Table 3 appears missing.
>
> We apologize for not stating this clearly in the paper. Due to space constraints, this information was originally included in **Table 5 in Appendix E.1**.
>
> ---
> ### References
> [1] Progress Measures for Grokking via Mechanistic Interpretability. ICLR'23.
>
> [2] Interpreting CLIP's Image Representation via Text-Based Decomposition. ICLR’24.
>
> [3] CoT-VLA: Visual Chain-of-Thought Reasoning for Vision-Language-Action Models. CVPR’25.

---

### Author Response · Authors · 2025-11-30
**Follow-up Summary for the Area Chair**

Dear AC,

We sincerely appreciate your tremendous efforts and valuable time in handling our submission. We are truly grateful for your guidance throughout this evaluation cycle. Our rebuttal and updated paper address all raised concerns, with major improvements across methodological soundness, empirical validation, and presentation clarity. Below is a concise summary of the key revisions:

- **Theoretical grounding of world modeling(Section 3.1)**: Leveraging the well-established theory that bounded-regret agents must implicitly model transition dynamics, we clarify the theoretical foundation behind our method.
- **Layer-wise zero-ablation study (Appendix E.4)**: We empirically show that deep layers play critical roles in VLA models, confirming that representation degradation harms the performance of VLAs, making correction necessary.
- **Expanded world-modeling baselines (Appendix E.5)**: We implement three baselines and conduct additional experiments to show that (i) SWOL outperforms these baselines, (ii) the performance gain arises from both mitigating the representation degradation and the integration of world modeling.
- **Real-world robustness (Appendix E.6)**: We add new real-world experiments on visual perturbation settings, demonstrating the visual robustness of SWOL.
- **Representation degradation analysis (Appendix E.7)**: Our empirical analysis shows that robot-data pretraining reduces semantic generalization, while enhancing dynamics generalization in the pre-trained VLM, and worsens degradation after finetuning.
- **RL validation (Appendix E.8)**: We conduct RL experiments for VLA models and demonstrate both the applicability and the superior performance of SWOL.
- **More ablations on the choice of target layers**: SWOL works robustly with (i) fixed mid-layer (e.g., layer 9), (ii) random layer (5–9), or (iii) ensemble mean (5–9 or 6–10), enabling plug-and-play use and offering a more stable target layer choice protocol.
- **Visualization of representation structure (Appendix E.9)**: t-SNE visualizations show that SWOL enables clearer separation among text, third-person, and gripper-view modalities. Quantitatively, SWOL increases action representation diversity.
- **Presentation improvements**: We (i) redesign Figure 1 and revise Section 2.1 to clarify perceptual/action token partitioning, (ii) revise Section 3.1 to emphasize world modeling as a theoretical necessity, (iii) add a discussion in Section 5 contrasting SWOL with visual representation learning methods  (iv) add a reproducibility statement in Appendix B, and (v) clarify the source of increased training cost in Appendix F.
----
## Remarks

Overall, we believe the revised paper now provides comprehensive evidence, resolves all reviewer concerns, and fairly reflects the contribution of our work. We would like to express our gratitude once again for your dedicated handling of our paper. We sincerely appreciate the time and careful consideration you’ve devoted to reviewing our work.

Best regards,

The Authors

---

### Meta-Review · Area_Chair_4WDc · 2026-01-08

**Summary:**

The paper investigates the phenomenon of representation degradation in Vision-Language-Action (VLA) models, demonstrating that deep layers often lose task-specific semantic and dynamic information. To mitigate this, the authors propose hidden Space WOrld modeLing (SWOL), an auxiliary self-supervised objective that aligns these degraded deep-layer features with the mid-layer representations of future observations using a lightweight MLP predictor. The method is shown to improve policy generalization on CALVIN and real-world robotic tasks without adding any inference-time overhead.

Overall, this paper received borderline scores. On one hand, reviewers acknowledged the performance gains and found the results to be interesting. On the other hand, major concerns were raised regarding the connection between the representation degradation measure and the policy, as well as the actual necessity of correcting this degradation. As explored in the study "Layer by Layer: Uncovering Hidden Representations in Language Models," deep representations in LLMs may not always be the most discriminative; however, this lack of discriminative power does not inherently necessitate improvement for the model to function effectively. After reviewing the manuscript and reviewer comments, these concerns remain for the following reasons: (1) the additional ablation results in Figure 10 show that ablating shallow layers has a smaller effect on performance than ablating deep layers. This contradicts the proposed representation degradation measure, which suggests that shallow layers possess "better" (more semantically rich) representations and should therefore be more critical to the task. (2) while the proposed method successfully mitigates some degradation in Figure 6, it does not fully resolve the issue, as the representation measures still show a distinct decrease from shallow to deep layers. Thus, I am inclined to agree with the reviewers that the correlation between the proposed representation degradation measure and actual policy performance requires more rigorous investigation.

**Reviewer Concerns:**

Some concerns, such as the comparison to prior world-modeling baselines, were addressed during the rebuttal by adding results for Direct World Modeling (DWM) and Direct World Modeling-Deep (DWMD). But note that the performance improvements offered by the proposed SWOL over these baselines appear marginal.

Concerns for the necessity of correcting the identified "representation degradation" remains questionable.  To address the connection between the representation degradation measure and policy performance, the authors provided a new ablation study in Figure 10. However, these results show that ablating shallow layers has a smaller impact than ablating deep layers. This finding seems contradict the authors' representation degradation measure, which suggests that shallow layers possess more semantically rich representations and should, therefore, be more critical. The ablation study indicates that the deep layers are actually more vital to policy performance. This discrepancy raises concerns regarding the validity and interpretation of the proposed degradation measure as a proxy for model effectiveness.

Additionally, the choice to align deep layers specifically with the mid-layers of future observations (also the fact that mid-layer quality varies significantly between different VLAs) remains largely empirical and lacks a theoretical or principled justification.

**Reviewer Scores:**

There was no further discussion after the rebuttal was posted. Had the reviewers been able to participate fully in the discussion, they might have adjusted their scores based on the authors' further clarifications. However, given the current state of the rebuttal, it is equally likely that they would have maintained their original scores.

---

### Decision · Program_Chairs · 2026-01-26

Reject